# Dynamic Fractal Mamba: A Neural Renormalization Group Flow for Scale-Invariant Sequence Modeling

Shenglei Fang [1]  Xianfang Sun [2]  You Zhou [1]

## Abstract

Sequence models typically operate at a fixed temporal or spatial scale and struggle to generalize to substantially longer horizons or higher resolutions without retraining. Existing hierarchical architectures expand receptive fields but rely on scale-specific parameters and lack mechanisms to enforce consistent dynamics across scales. We propose **Dynamic Fractal Mamba (DF-Mamba)**, a recursive state-space model that applies a single shared operator across multiple scales. By sharing parameters across recursion depths and exponentially scaling the effective time step, DF-Mamba achieves an exponentially expanding receptive field while preserving linear computational complexity. A learned content-aware coarse-graining module aggregates representations across scales. Auxiliary reconstruction and cross-scale consistency objectives stabilize recursive training. We evaluate DF-Mamba on long-range time-series forecasting, spatial transcriptomics, and computational pathology. Across all tasks, DF-Mamba consistently outperforms Transformers and flat Mamba baselines while using fewer parameters and maintaining linear-time scalability. Importantly, models trained on short sequences or low-resolution inputs generalize in a zero-shot manner to substantially larger temporal and spatial scales unseen during training. These results demonstrate that recursive parameter sharing provides an effective inductive bias for learning scale-consistent and efficient sequence representations. Our code is available at: https://github.com/yzlab1/Dynamic-Fractal-Mamba

[1]Department of Infection and Immunity, School of Medicine, Cardiff University, Cardiff, UK [2]School of Computer Science and Informatics, Cardiff University, Cardiff, UK. Correspondence to: You Zhou <zhouy58@cardiff.ac.uk>.

*Proceedings of the 43$^{rd}$ International Conference on Machine Learning*, Seoul, South Korea. PMLR 306, 2026. Copyright 2026 by the author(s).

## 1. Introduction

Modern sequence models have achieved impressive performance across natural language, vision, and scientific data by scaling depth, context length, and model capacity. However, most architectures implicitly overfit to the temporal or spatial scales observed during training (Bassingthwaighte et al., 1994; Porcaro et al., 2024; Havlin et al., 1995; Mageed, 2025). When evaluated at substantially larger resolutions or longer horizons, performance often degrades sharply without retraining or architectural modification. For example, models trained on short temporal windows frequently fail to extrapolate to orders-of-magnitude longer sequences, and patch-level vision models trained on local crops struggle to generalize reliably to whole-slide pathology images. This lack of *scale generalization* limits the deployment of sequence models in scientific and medical settings, where inference routinely occurs far beyond the training scale.

Existing architectures offer limited remedies. Transformers suffer from prohibitive quadratic complexity on ultra-long sequences (Vaswani et al., 2017). While linear-time SSMs like Mamba improve scalability (Gu & Dao, 2024; Dao & Gu, 2024), their flat hierarchy struggles to disentangle multi-scale dynamics, causing capacity bottlenecks (Chen et al., 2024; Zheng et al., 2025). Furthermore, standard hierarchical models typically rely on disjoint, scale-specific parameters without enforcing cross-scale consistency, which restricts extrapolation beyond the training regime.

We argue that effective scale generalization requires *recursive parameter sharing*: the same dynamics should be applied repeatedly across resolutions so that representations evolve consistently as scale increases. Similar principles have been explored in the context of renormalization in statistical physics (Machado & Saueressig, 2008; Komargodski & Schwimmer, 2011; Koch et al., 2019). This principle enables exponential growth of the receptive field while keeping the number of learned parameters fixed and computational cost linear in sequence length.

Motivated by this observation, we propose **Dynamic Fractal Mamba (DF-Mamba)**, a recursive state-space sequence model that applies a single shared operator across multiple scales. DF-Mamba combines local state-space model-

ing with a learned content-aware coarse-graining module that progressively aggregates information across resolutions. Recursive application of the shared operator expands the receptive field exponentially while preserving linear computational complexity. To stabilize recursive learning, we introduce auxiliary reconstruction and cross-scale consistency objectives that encourage alignment across recursion depths (Shiina et al., 2021). We evaluate DF-Mamba on three complementary domains: long-range time-series forecasting, spatial transcriptomics, and computational pathology. Across all tasks, DF-Mamba consistently outperforms Transformers and flat Mamba baselines while using fewer parameters and maintaining linear-time scalability. Importantly, models trained on short sequences or low-resolution inputs generalize in a zero-shot manner to substantially larger temporal and spatial scales unseen during training. These results demonstrate that recursive parameter sharing provides an effective inductive bias for learning scale-consistent and efficient sequence representations.

In summary, this work makes the following contributions:

- We introduce DF-Mamba, a recursive state-space architecture that applies a single shared operator across multiple scales to achieve exponential receptive field growth with linear computational complexity.

- We propose a learned content-aware coarse-graining mechanism and auxiliary reconstruction and cross-scale consistency objectives that stabilize recursive training.

- We demonstrate strong empirical performance and zero-shot scale extrapolation across long-range time-series forecasting, spatial transcriptomics, and computational pathology benchmarks.

## 2. Related Work

### 2.1. Efficient Sequence Modeling

Modeling long-range dependencies is central to deep learning. Transformers have dominated this landscape but suffer from quadratic complexity $O(L^2)$, limiting their applicability to ultra-long sequences (Vaswani et al., 2017). Efficient variants attempt to reduce this cost but often compromise expressivity (Shen et al., 2021; Kitaev et al., 2020; Ma et al., 2021; Acharya et al., 2024; Zhao et al., 2025). Recently, Structured State Space Models (SSMs), such as S4 and Mamba, have emerged as powerful alternatives, offering linear time complexity $O(L)$ and parallel training (Gu et al., 2021; Gu & Dao, 2024; Dao & Gu, 2024). In computer vision, adaptations like Vim and VMamba process images via flat 2D scanning (Zhu et al., 2024; Liu et al., 2024). However, these standard SSMs typically operate on a flat hierarchy. They process tokens sequentially without explicitly

modeling the multi-scale structure inherent in physical data, often struggling to distinguish high-frequency noise from low-frequency macroscopic trends in complex scenarios.

### 2.2. Renormalization Group and Dynamic Fractal

The Renormalization Group (RG) framework in statistical physics provides a rigorous method for analyzing changes in a system across scales (Kadanoff, 1966; Wilson, 1971a;b). Specifically, RG flow drives a system towards critical fixed points where the governing dynamics become scale-invariant. This emergence of statistical self-similarity, where macroscopic structures mirror microscopic details, constitutes an intrinsic fractal geometry (Gao et al., 2019). The intersection of Renormalization Group (RG) theory and deep learning is a significant research frontier, moving from qualitative analogies to rigorous mathematical frameworks that enhance model stability, interpretability, and generalization (Lee et al., 2025; Xia & Gong, 2025). However, current RG-deep learning frameworks typically rely on fixed, handcrafted transformations or static hierarchical architectures. Bridging this gap, DF-Mamba operationalizes the dynamic fractal inductive bias of RG flow within the continuous dynamics of State Space Models. While DF-Mamba shares structural parallels with hierarchical backbones (such as U-Net(Ronneberger et al., 2015) and Swin Transformer (Liu et al., 2021)), it fundamentally differs in mechanism. In contrast, DF-Mamba enforces strictly shared dynamics across scales, constrained by the renormalization group flow, thereby learning a universal physical operator rather than scale-specific approximations. DF-Mamba is constraint-driven: our exponential time-scaling ($\Delta_k$) and recursive weight sharing are mathematically necessitated by RG flow consistency. This constraint-driven design enables learning a reusable operator across resolutions and underlies the zero-shot scale extrapolation observed in our experiments.

## 3. Method

We introduce **DF-Mamba**, a recursive state-space architecture that operationalizes the renormalization group flow to achieve scale-invariant sequence modeling. Unlike pyramidal or U-Net-style backbones that learn scale-specific parameters at each resolution, DF-Mamba enforces a scale-invariant design by reusing the same parameterized state-space operator $\Phi_\theta$ across all recursion levels. This enables exponential growth of the effective receptive field without increasing the number of learned parameters, while keeping computation linear in sequence length. The model combines local state-space modeling with a learned coarse-graining module and gated multi-scale feature fusion to support stable recursive computation and generalization under scale shifts.

### 3.1. State Space Model Backbone

DF-Mamba builds upon the Selective State Space Model (SSM) framework, which provides linear-time sequence modeling with strong inductive bias for long-range dependencies. An SSM maps a 1D input sequence $x(t) \in \mathbb{R}$ to an output $y(t) \in \mathbb{R}$ through a latent state $h(t) \in \mathbb{R}^N$. The continuous-time dynamics are governed by

$$\frac{dh(t)}{dt} = \mathbf{A}_c h(t) + \mathbf{B}_c x(t), \qquad y(t) = \mathbf{C}_c h(t), \quad (1)$$

where $\mathbf{A}_c \in \mathbb{R}^{N \times N}$, $\mathbf{B}_c \in \mathbb{R}^{N \times 1}$, and $\mathbf{C}_c \in \mathbb{R}^{1 \times N}$ are time-invariant parameters of the continuous dynamics.

Under the Zero-Order Hold (ZOH) assumption with a discretization step $\Delta > 0$, the system is discretized as

$$h_{t+1} = \overline{\mathbf{A}}\, h_t + \overline{\mathbf{B}}\, x_t, \qquad y_t = \mathbf{C} h_t, \qquad where \mathbf{C} = \mathbf{C}_c \tag{2}$$

with

$$\overline{\mathbf{A}} = \exp(\Delta \mathbf{A}_c), \qquad \overline{\mathbf{B}} = \left( \int_0^\Delta \exp(\tau \mathbf{A}_c)\, d\tau \right) \mathbf{B}_c. \tag{3}$$

When $\mathbf{A}_c$ is invertible, $\overline{\mathbf{B}}$ admits the closed form

$$\overline{\mathbf{B}} = \mathbf{A}_c^{-1} \big( \exp(\Delta \mathbf{A}_c) - \mathbf{I} \big) \mathbf{B}_c, \tag{4}$$

In practice, $\overline{\mathbf{A}}$ and $\overline{\mathbf{B}}$ are computed using stable matrix exponential routines without explicitly forming matrix inverses. Following prior work, we parameterize the continuous dynamics using a diagonal structure

$$\mathbf{A}_c = \mathrm{diag}(-\exp(\mathbf{A}_{\log})),$$

which ensures non-positive eigenvalues and stable state transitions. This parameterization supports large discretization steps without numerical instability and enables efficient parallel computation across channels.

In DF-Mamba, the discretization step $\Delta$ is treated as a content-adaptive parameter that controls the effective temporal receptive field. In the recursive architecture described next, $\Delta$ is further scaled as a function of recursion depth to enable exponential expansion of the receptive field while preserving linear computational complexity.

### 3.2. The Recursive RG-Flow Architecture

To capture the dynamic fractal nature of sequential data, DF-Mamba abandons the flat structure of traditional Mamba blocks. Instead, we define a recursive operator $\Phi_\theta$ (corresponds to the generic "Level $k$" module (where $k = 0, 1, \ldots$) in Figure 1) that processes the input $\mathbf{X}_k$ at depth $k$ (scale $2^k$).

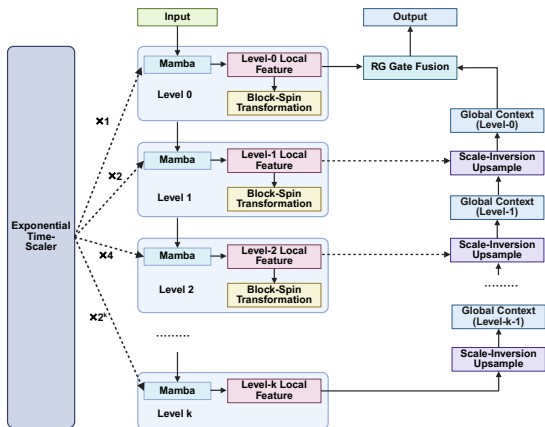

*Figure 1.* The overall architecture of Dynamic Fractal Mamba (DF-Mamba).

#### 3.2.1. EXPONENTIAL TIME-STEP SCALING

To efficiently integrate information over long temporal ranges, the discretization step size is scaled as a function of recursion depth. For input features $\mathbf{X}_k$ at depth $k$, the effective step size is defined as

$$\Delta_k(\mathbf{X}_k) = \mathrm{Softplus}(\mathrm{Linear}(\mathbf{X}_k)) \cdot 2^k. \tag{5}$$

The content-adaptive term follows the standard Mamba parameterization, while the multiplicative factor $2^k$ **explicitly models the physical time dilation** $(dt \to 2^k dt)$ required by RG flow consistency, ensuring the receptive field expands exponentially.

To ensure numerical stability under large step sizes, the continuous dynamics are parameterized using a diagonal matrix

$$\mathbf{A}_c = \mathrm{diag}(-\exp(\mathbf{A}_{\log})),$$

which guarantees non-positive eigenvalues and contractive state transitions. In practice, $\Delta_k$ is clipped within a safe range to prevent excessive damping at large recursion depths.

#### 3.2.2. LEARNED COARSE-GRAINING

To simulate the physical block-spin transformation, we introduce a learnable coarse-graining operator $\mathcal{D} : \mathbb{R}^{L \times D} \to \mathbb{R}^{\lceil L/2 \rceil \times D}$ that aggregates interactions between adjacent token pairs. Concretely, we form non-overlapping blocks by concatenating neighbors:

$$\mathbf{P}_k = \mathrm{PairConcat}(\mathbf{H}_{\mathrm{local}}^{(k)}) \in \mathbb{R}^{\lceil L/2 \rceil \times 2D}, \tag{6}$$

$$\mathbf{X}_{k+1} = \mathrm{MLP}_{enc}(\mathbf{P}_k) \in \mathbb{R}^{\lceil L/2 \rceil \times D}, \tag{7}$$

where PairConcat handles odd sequence lengths $L$ by zero-padding the final block, and $\mathbf{H}_{\text{local}}^{(k)}$ denotes the hidden states from the Mamba block at recursion depth $k$. The encoder $\mathcal{D}$ is parameterized as a lightweight two-layer MLP with GELU activation, applied independently to each token pair.

**Parameter Sharing Constraint.** Crucially, we *share* the parameters of $\text{MLP}_{enc}$ across all recursion depths $k$. This design implies that the coarse-graining rule is scale-invariant, explicitly enforcing scale-consistent renormalization dynamics regardless of the observational resolution.

**Information Conservation.** To ensure that relevant microscopic information is preserved during coarsening, we equip $\mathcal{D}$ with a symmetric decoder $\mathcal{D}^{\dagger}$, implemented as $\hat{\mathbf{P}}_k = \text{MLP}_{dec}(\mathbf{X}_{k+1})$, which is optimized via an auxiliary reconstruction loss $\mathcal{L}_{recon}$.

### 3.2.3. RECURSIVE INTEGRATION AND RG-GATED FUSION

The architecture is defined recursively. Let Local Feature $\mathbf{H}_{\text{local}}^{(k)}$ denote the features produced by the SSM at recursion depth $k$. The global context $\mathbf{H}_{\text{global}}^{(k)}$ is obtained by recursively calling the model on the coarse-grained input $\mathbf{X}_{k+1}$ and upsampling:

$$\mathbf{H}_{\text{global}}^{(k)} = \mathcal{U}\left(\Phi_\theta(\mathbf{X}_{k+1}, k+1)\right), \tag{8}$$

where $\mathcal{U}$ is a linear interpolation upsampler, and $\Phi_\theta(\cdot, k+1)$ denotes the recursive execution of the DF-Mamba network at the next coarser scale parameterized by weights $\theta$. This recursive call retrieves the macroscopic representation from deeper levels of the fractal hierarchy.

Finally, a **Renormalization Group Gate** $\mathbf{g} \in (0, 1)$ fuses the microscopic and macroscopic views based on local information density:

$$\mathbf{H}_{\text{out}}^{(k)} = (1 - \mathbf{g}) \odot \mathbf{H}_{\text{local}}^{(k)} + \mathbf{g} \odot \mathbf{H}_{\text{global}}^{(k)}. \tag{9}$$

This gate allows the model to dynamically switch between detailed local processing and global trend aggregation.

## 3.3. Physics-Informed Constraints

Training deep recursive networks is notoriously unstable due to vanishing gradients and semantic drift. To stabilize learning and enforce physical plausibility, we introduce two auxiliary objectives derived from RG principles.

**Information Conservation Constraint ($\mathcal{L}_{recon}$).** In RG flow, coarse-graining should preserve relevant operators. We treat downsampling as a learnable compression step and enforce information preservation via reconstruction in the blocked space. Specifically, local features $\mathbf{H}_{\text{local}}^{(k)}$ are reshaped into non-overlapping blocks

$$\mathbf{P}_k = \text{PAIRCONCAT}(\mathbf{H}_{\text{local}}^{(k)}) \in \mathbb{R}^{\lceil L/2 \rceil \times 2D}, \tag{10}$$

and coarse-grained by $\mathbf{X}_{k+1} = \mathcal{D}(\mathbf{P}_k)$. Equipping $\mathcal{D}$ with a decoder $\mathcal{D}^{\dagger}$, we minimize

$$\begin{aligned}
\mathcal{L}_{recon} &= \sum_{k=0}^{K-1} \left\| \text{sg}(\mathbf{P}_k) - \mathcal{D}^{\dagger}(\mathbf{X}_{k+1}) \right\|_2^2 \\
&= \sum_{k=0}^{K-1} \left\| \text{sg}(\mathbf{P}_k) - \mathcal{D}^{\dagger}(\mathcal{D}(\mathbf{P}_k)) \right\|_2^2,
\end{aligned} \tag{11}$$

where $\text{sg}(\cdot)$ denotes stop-gradient. This encourages the coarse-grained representation $\mathbf{X}_{k+1}$ to retain fine-scale information present in the pre-coarsening blocks $\mathbf{P}_k$.

**Scale-Invariant Consistency ($\mathcal{L}_{consist}$).** At a critical point (fixed point of the RG flow), the system exhibits self-similarity. Geometrically, this implies that the upsampled macroscopic representation should approximate the microscopic representation. We enforce this *Fixed Point Property* via a consistency loss:

$$\mathcal{L}_{consist} = \sum_{k=0}^{K-1} \|\text{sg}(\mathbf{H}_{local}^{(k)}) - \mathbf{H}_{global}^{(k)}\|_2^2 \tag{12}$$

where $\text{sg}(\cdot)$ denotes the stop-gradient operator. This prevents the global context from collapsing and forces the network to learn a unified physical law across scales.

**Total Objective.** The final loss is a weighted sum of the task loss and physical constraints: $\mathcal{L} = \mathcal{L}_{task} + \lambda_1 \mathcal{L}_{recon} + \lambda_2 \mathcal{L}_{consist} + \lambda_3 \mathcal{L}_{flow}$. Additionally, we apply a minor smoothness regularization $\mathcal{L}_{flow}$ on the gate values to stabilize recursive updates

## 3.4. Computational Complexity

Despite its recursive nature, DF-Mamba maintains linear complexity. Since the sequence length is halved at each recursion depth, the total computational cost forms a geometric series:

$$\begin{aligned}
\mathcal{C}_{\text{total}} &= \sum_{k=0}^{K} \mathcal{C}_{\text{SSM}}\left(\frac{L}{2^k}\right) \\
&= O(L)\left(1 + \frac{1}{2} + \frac{1}{4} + \cdots\right) \approx 2\,O(L).
\end{aligned} \tag{13}$$

Thus, DF-Mamba retains the $O(L)$ efficiency of standard Mambas while theoretically expanding the effective receptive field to infinity.

# 4. Experiment

We propose Dynamic Fractal Mamba (DF-Mamba). To test whether DF-Mamba learns representations stable under scale transformations, we evaluate it on three complementary domains, each targeting a key component of our

design. (i) Time series forecasting probes exponential time scaling by requiring simultaneous modeling of slow global trends and fast local fluctuations. (ii) Spatial transcriptomics (ST) tests block-spin downsampling in a 2D biological setting, where fine-grained expression must be aggregated into tissue-level structure without losing functional organization. (iii) Computational pathology classification stresses cross-scale consistency under weak supervision and large magnification shifts.

## 4.1. Experiments on Time Series Forecasting

**Data** To validate our model, we evaluate DF-Mamba on 4 benchmarks: 2 ETTs (ETTh2 and ETTm2), Electricity (ECL), and Weather (Wu et al., 2021). We trained and tested our model with 60/20/20 train/validation/test split for ETTs, and 70/10/20 for other datasets. We use the look-back window of 336 for ETTs, 512 for Weather, and 96 for Electricity, similar to (Wu et al., 2025) setting. All datasets are normalized during training.

**Baseline** We assessed DF-Mamba against eight baselines, including: Transformers (iTransformer(Liu et al., 2023),PatchTST (Nie, 2022), and Crossformer(Zhang & Yan, 2023)), CNNs (TSLANet(Eldele et al., 2024)), MLPS (Dlinear(Zeng et al., 2023) and Rlinear(Li et al., 2023)), and Mambas (DTMamba(Wu et al., 2024) and Affirm(Wu et al., 2025))

**Result** We evaluate DF-Mamba for time series Forecasting (DF-Mamba-TS) on four widely-used multivariate long-term forecasting benchmarks, including ETTh2, ETTm2, Weather, and Electricity, with prediction horizons of 96, 192, 336, and 720. Table 1 reports the MSE and MAE results across all settings.

Overall, DF-Mamba-TS demonstrates consistently strong performance across diverse datasets and forecasting horizons. On the challenging ETTh2 and ETTm2 datasets, DF-Mamba-TS achieves the best or second-best results in most configurations, with particularly clear advantages at longer prediction horizons (336 and 720). This indicates that the proposed model effectively captures long-range temporal dependencies while maintaining stable predictive accuracy.

Compared with attention-based methods, such as iTransformer and Crossformer, DF-Mamba-TS exhibits significantly improved robustness as the prediction horizon increases. Notably, Crossformer suffers from severe performance degradation in long-horizon forecasting, whereas DF-Mamba-TS maintains stable performance across all horizons. This behavior highlights the benefit of the linear-time state-space formulation in modeling long-term temporal dynamics.

On the Weather and Electricity datasets, DF-Mamba-TS remains competitive with strong Transformer-based and linear baselines, achieving comparable performance on short horizons and consistently strong results on longer horizons.

These results suggest that DF-Mamba-TS preserves short-term forecasting accuracy while improving stability for long-range prediction.

In summary, the results across all benchmarks validate the effectiveness of DF-Mamba-TS for long-term multivariate time-series forecasting, particularly in scenarios requiring robust modeling of long-range temporal dependencies.

**Ablation Study.** We conduct ablations on ETTm2 (Table 2) to isolate the contributions of architecture, time-scaling, mechanisms, and losses. DF-Mamba-TS achieves the best performance (MSE/MAE $0.158/0.261$) with $0.658$M parameters. In contrast, a standard stacked backbone (S2) degrades substantially ($0.210/0.288$), suggesting that simply stacking Mamba blocks fails to induce hierarchical long-range structure. Unrolling the recursion without weight sharing (S3) partially recovers accuracy ($0.180/0.282$) but increases parameters ($0.757$M vs. $0.658$M), indicating that shared dynamics provide both parameter efficiency and an implicit regularization toward scale-consistent temporal features. Crucially, unlike standard hierarchical models (such as U-Net-style encoder–decoder) that learn disjoint, scale-specific parameters, DF-Mamba captures a universal renormalization operator across resolutions, enabling superior generalization.

Exponential time scaling is also critical: replacing $\Delta_k \propto 2^k$ with linear (T2) or constant (T3) scaling worsens MSE to $0.194$ and $0.186$, respectively, consistent with reduced effective receptive-field growth at deeper recursion. For coarse-graining and fusion, naive pooling (C2) increases MSE to $0.169$, while replacing gated fusion with simple addition (C3) yields $0.163$ (vs. $0.158$), indicating that learnable downsampling better preserves informative high-frequency content and gating resolves conflicts between local details and global trends. Finally, auxiliary constraints are necessary for stable training: using only the task loss (L4) drops performance to MSE $0.174$, and removing consistency (L2: $0.167$) or reconstruction (L3: $0.171$) also degrades results, showing that these constraints guide optimization toward scale-aligned representations.

## 4.2. Experiments on Spatial Transcriptomics

**Data** We evaluate the performance of our model on the widely-used human dorsolateral prefrontal cortex (DLPFC) dataset from the Lieber Institute for Brain Development (LIBD) (Maynard et al., 2021). This benchmark dataset contains 12 spatially-resolved transcriptomics slices, each with expert-annotated ground-truth labels corresponding to the four or six distinct cortical layers and white matter (WM). Its well-organized, multi-layered tissue architecture provides

*Table 1.* Multivariate long-term series forecasting results on different prediction lengths $\in \{96, 192, 336, 720\}$. The best results are highlighted in **bold**, and the second-best results are underlined.

| METHODS | DF-MAMBA-TS | | AFFIRM | | DTMAMBA | | TSLANET | | iTRANSFORMER | | PATCHTST | | CROSSFORMER | | RLINEAR | | DLINEAR | |
|---|---|---|---|---|---|---|---|---|---|---|---|---|---|---|---|---|---|---|
| METRICS | MSE | MAE | MSE | MAE | MSE | MAE | MSE | MAE | MSE | MAE | MSE | MAE | MSE | MAE | MSE | MAE | MSE | MAE |
| ETTh2 96 | **0.232** | **0.323** | 0.276 | 0.343 | 0.290 | 0.340 | 0.280 | 0.341 | 0.297 | 0.349 | 0.285 | 0.340 | 0.745 | 0.584 | 0.288 | 0.338 | 0.289 | 0.353 |
| ETTh2 192 | **0.287** | **0.364** | 0.316 | 0.372 | 0.366 | 0.392 | 0.330 | 0.375 | 0.380 | 0.400 | 0.356 | 0.386 | 0.877 | 0.656 | 0.374 | 0.390 | 0.383 | 0.418 |
| ETTh2 336 | 0.329 | 0.393 | 0.341 | 0.383 | 0.380 | 0.409 | **0.317** | **0.374** | 0.428 | 0.432 | 0.350 | 0.395 | 1.043 | 0.731 | 0.415 | 0.426 | 0.448 | 0.465 |
| ETTh2 720 | 0.408 | 0.437 | **0.391** | 0.429 | 0.416 | 0.437 | 0.404 | 0.440 | 0.427 | 0.445 | 0.395 | **0.427** | 1.104 | 0.763 | 0.420 | 0.440 | 0.605 | 0.551 |
| ETTm2 96 | **0.158** | 0.261 | 0.167 | 0.260 | 0.177 | 0.259 | 0.169 | 0.259 | 0.180 | 0.264 | 0.169 | **0.254** | 0.287 | 0.366 | 0.182 | 0.265 | 0.167 | 0.260 |
| ETTm2 192 | **0.193** | **0.293** | 0.221 | 0.296 | 0.240 | 0.300 | 0.224 | 0.297 | 0.250 | 0.309 | 0.230 | 0.294 | 0.414 | 0.492 | 0.246 | 0.304 | 0.224 | 0.303 |
| ETTm2 336 | **0.239** | **0.328** | 0.271 | **0.328** | 0.310 | 0.345 | 0.275 | 0.329 | 0.311 | 0.348 | 0.280 | 0.329 | 0.597 | 0.542 | 0.307 | 0.342 | 0.281 | 0.342 |
| ETTm2 720 | **0.316** | **0.375** | 0.351 | 0.377 | 0.395 | 0.394 | 0.354 | 0.380 | 0.412 | 0.407 | 0.378 | 0.386 | 1.730 | 1.042 | 0.407 | 0.398 | 0.397 | 0.421 |
| WEATHER 96 | 0.149 | **0.194** | **0.146** | 0.196 | 0.171 | 0.218 | 0.148 | 0.197 | 0.174 | 0.214 | 0.160 | 0.204 | 0.158 | 0.230 | 0.192 | 0.232 | 0.176 | 0.237 |
| WEATHER 192 | 0.193 | 0.242 | **0.192** | **0.239** | 0.220 | 0.257 | 0.193 | 0.241 | 0.221 | 0.254 | 0.204 | 0.245 | 0.206 | 0.277 | 0.240 | 0.271 | 0.220 | 0.282 |
| WEATHER 336 | **0.237** | 0.264 | 0.244 | 0.278 | 0.274 | 0.296 | 0.245 | 0.282 | 0.278 | 0.296 | 0.257 | 0.285 | 0.272 | 0.335 | 0.292 | 0.307 | 0.265 | 0.319 |
| WEATHER 720 | 0.319 | 0.330 | 0.321 | 0.332 | 0.349 | 0.346 | 0.325 | 0.337 | 0.358 | 0.349 | 0.329 | 0.338 | **0.298** | 0.418 | 0.364 | 0.353 | 0.323 | 0.362 |
| ELECTR. 96 | 0.133 | 0.268 | **0.129** | **0.223** | 0.166 | 0.256 | 0.136 | 0.229 | 0.148 | 0.240 | 0.138 | 0.230 | 0.219 | 0.314 | 0.201 | 0.281 | 0.140 | 0.237 |
| ELECTR. 192 | 0.154 | 0.279 | **0.146** | **0.239** | 0.178 | 0.268 | 0.152 | 0.244 | 0.162 | 0.253 | 0.149 | 0.243 | 0.231 | 0.322 | 0.201 | 0.283 | 0.153 | 0.249 |
| ELECTR. 336 | **0.157** | 0.282 | 0.162 | **0.252** | 0.197 | 0.289 | 0.168 | 0.262 | 0.178 | 0.269 | 0.169 | 0.262 | 0.246 | 0.337 | 0.215 | 0.298 | 0.169 | 0.267 |
| ELECTR. 720 | 0.200 | 0.317 | **0.191** | **0.288** | 0.243 | 0.326 | 0.205 | 0.293 | 0.225 | 0.317 | 0.211 | 0.299 | 0.280 | 0.363 | 0.257 | 0.331 | 0.203 | 0.301 |

*Table 2.* Ablation study results of DF-Mamba-TS and its variants on the ETTm2 dataset.

| Model | MSE | MAE | Params (M) | FLOPs (G) | INF. (s) |
|---|---|---|---|---|---|
| DF-Mamba-TS | 0.158 | 0.261 | 0.658 | 0.057 | 9.666 |
| S2 | 0.210 | 0.288 | 0.699 | 0.066 | 12.108 |
| S3 | 0.180 | 0.282 | 0.757 | 0.057 | 10.059 |
| T2 | 0.194 | 0.288 | 0.658 | 0.057 | 9.943 |
| T3 | 0.186 | 0.284 | 0.658 | 0.057 | 9.847 |
| C2 | 0.169 | 0.274 | 0.641 | 0.052 | 9.612 |
| C3 | 0.163 | 0.261 | 0.649 | 0.048 | 9.570 |
| L2 | 0.167 | 0.270 | 0.658 | 0.057 | 9.593 |
| L3 | 0.171 | 0.274 | 0.658 | 0.057 | 9.564 |
| L4 | 0.174 | 0.270 | 0.658 | 0.057 | 10.270 |

an ideal testbed for evaluating a model's capacity to learn representations that preserve spatially coherent biological structures. In line with standard practice, we log-normalize the raw gene expression counts and use the top 3,000 highly variable genes (HVGs) as input features. To demonstrate our model is not tailored to DLPFC only, we additionally evaluate on the 10x Genomics human breast cancer Visium dataset (Cui et al., 2025).

**Baseline** The task is unsupervised spatial clustering to recover annotated anatomical layers. We benchmark our DF-Mamba model for spatial transcriptomics (DF-Mamba-ST) against six state-of-the-art baselines: BayesSpace(Zhao et al., 2021), SpaGCN(Hu et al., 2021), DeepST(Xu et al., 2022), GraphST(Long et al., 2023), BASS(Li & Zhou, 2022), and DiffusionST(Cui et al., 2025). Baselines were executed using their officially suggested hyperparameters for a fair comparison. We quantify clustering accuracy using two standard metrics: the Adjusted Rand Index (ARI) and Normalized Mutual Information (NMI) are reported in the main paper as the primary performance metric.

**Result** We further evaluate DF-Mamba on spatial transcriptomics datasets (DF-Mamba-ST), including the DLPFC benchmark with 12 tissue slices and the Breast Cancer dataset. Table 3 reports clustering performance measured by ARI and NMI.

On the DLPFC dataset, DF-Mamba-ST achieves the highest average ARI (0.51) and competitive NMI (0.63), outperforming all baseline methods. Notably, DF-Mamba-ST maintains robust performance across all tissue sections, particularly on challenging samples (such as 151672 and 151674) where other graph-based or diffusion-based methods often struggle. This indicates our model's superior capability in handling varying spatial patterns and tissue structures.

On the Breast Cancer dataset, DF-Mamba-ST demonstrates a substantial performance leap, achieving an ARI of 0.65 and an NMI of 0.69. It surpasses the second-best method (DiffusionST) by significant margins of 14% in ARI and nearly 28% in NMI. These improvements highlight the advantage of our recursive fractal modeling, which effectively captures the complex, multi-scale heterogeneity typical of tumor microenvironments without relying on explicit graph construction or computationally expensive diffusion processes.

Overall, these results suggest that DF-Mamba-ST effectively models multi-scale spatial dependencies, offering a robust and efficient alternative to traditional graph-based approaches for spatial domain identification.

### 4.3. Experiments on Computational Pathology

Beyond standard in-distribution classification, we use computational pathology as a stress test for cross-scale generalization. In practical histopathology, models are often trained with limited local fields-of-view (due to weak supervision or sparse annotations), while diagnostic decisions depend

*Table 3.* Comparison of clustering performance (ARI and NMI) on DLPFC and Breast Cancer datasets against SOTA methods. The best results are highlighted in **bold**, and the second-best results are underlined.

| | Methods | BayesSpace | | SpaGCN | | DeepST | | GraphST | | BASS | | DiffusionST | | DF-Mamba-ST | |
|---|---|---|---|---|---|---|---|---|---|---|---|---|---|---|---|
| | Metrics | ARI | NMI | ARI | NMI | ARI | NMI | ARI | NMI | ARI | NMI | ARI | NMI | ARI | NMI |
| DLPFC | 151507 | 0.47 | 0.59 | 0.46 | 0.54 | 0.50 | 0.62 | 0.51 | **0.65** | 0.50 | 0.63 | 0.43 | 0.64 | **0.52** | 0.64 |
| | 151508 | 0.44 | 0.56 | 0.43 | 0.52 | 0.39 | 0.54 | 0.35 | 0.57 | 0.46 | 0.57 | 0.46 | **0.59** | **0.47** | 0.58 |
| | 151509 | 0.40 | 0.55 | 0.47 | **0.62** | 0.36 | 0.57 | 0.41 | 0.59 | 0.51 | **0.62** | **0.54** | **0.62** | 0.53 | **0.62** |
| | 151510 | 0.38 | 0.52 | 0.46 | 0.54 | 0.38 | 0.55 | 0.41 | **0.61** | 0.50 | 0.58 | **0.51** | **0.61** | **0.51** | **0.61** |
| | 151669 | **0.47** | **0.59** | 0.37 | 0.52 | 0.35 | 0.51 | 0.27 | 0.50 | 0.35 | 0.52 | 0.38 | 0.58 | 0.39 | 0.56 |
| | 151670 | 0.43 | 0.52 | 0.38 | 0.49 | 0.34 | 0.48 | 0.25 | 0.46 | 0.38 | 0.51 | **0.46** | **0.57** | 0.45 | 0.55 |
| | 151671 | 0.49 | 0.62 | 0.56 | 0.68 | 0.52 | 0.69 | 0.50 | 0.67 | 0.56 | 0.71 | **0.58** | **0.72** | 0.57 | 0.69 |
| | 151672 | 0.44 | 0.58 | 0.57 | 0.68 | 0.50 | 0.67 | 0.58 | **0.69** | 0.56 | 0.65 | 0.55 | 0.66 | **0.59** | **0.69** |
| | 151673 | **0.55** | 0.69 | 0.52 | 0.51 | **0.55** | 0.68 | 0.51 | 0.66 | 0.53 | 0.72 | 0.54 | **0.73** | 0.53 | 0.69 |
| | 151674 | 0.32 | 0.48 | 0.39 | 0.53 | 0.49 | 0.63 | 0.48 | 0.61 | 0.51 | 0.62 | **0.52** | **0.69** | 0.51 | 0.68 |
| | 151675 | 0.53 | 0.69 | 0.47 | 0.58 | **0.57** | 0.72 | 0.52 | 0.64 | 0.55 | **0.74** | 0.46 | 0.61 | 0.53 | 0.64 |
| | 151676 | 0.37 | 0.53 | 0.33 | 0.48 | 0.52 | 0.59 | 0.42 | 0.56 | **0.53** | **0.70** | 0.51 | 0.68 | 0.49 | 0.62 |
| | DLPFC Avg. | 0.44 | 0.58 | 0.45 | 0.56 | 0.46 | 0.60 | 0.43 | 0.60 | 0.50 | 0.63 | 0.50 | **0.64** | **0.51** | 0.63 |
| | Breast Cancer | 0.49 | 0.53 | 0.57 | 0.61 | 0.51 | 0.52 | 0.53 | 0.54 | 0.55 | 0.54 | 0.57 | 0.54 | **0.65** | **0.69** |

on global tissue organization at substantially larger spatial extents. Therefore, the key question we evaluate is whether a model can extrapolate from local training views to unseen larger resolutions without retraining or scale-specific adaptation.

**Zero-shot Scale Extrapolation.** A model exhibits zero-shot scale extrapolation if it is trained exclusively on inputs at a source scale $S_{\text{train}}$ (such as local crops or short contexts) and evaluated at a strictly larger target scale $S_{\text{test}} > S_{\text{train}}$ (such as larger regions or longer contexts), without architectural modification, scale-specific fine-tuning, or exposure to $S_{\text{test}}$ during training. In this setting, performance at $S_{\text{test}}$ reflects the model's ability to maintain scale-consistent representations rather than memorizing resolution-specific textures.

**Data** We evaluate DF-Mamba on the breast cancer histopathology dataset (Aresta et al., 2019), which consists of 500 microscopy images annotated at the image level by two expert pathologists from the Institute of Molecular Pathology and Immunology of the University of Porto (IPATIMUP) and the Institute for Research and Innovation in Health. Following standard practice, we split the dataset into 70%/15%/15% for training, validation, and testing, respectively.

During training, all images are first resized to a unified resolution of $512 \times 512$. To simulate limited local context and reduce annotation dependency, the model is trained exclusively on randomly cropped local crops of size $S \times S$, where $S \in 128, 256$. These crops correspond to token sequence lengths of $L_{train} = 64$ and $L_{train} = 256$, respectively, given a fixed patch size of $16 \times 16$. Random cropping and spatial augmentations (horizontal and vertical flips) are applied to improve robustness.

At validation and test time, the trained model is evaluated on the full-resolution $512 \times 512$ images without any cropping, resulting in a substantially longer token sequence of length $L_{test} = 1024$. Importantly, this global resolution is never observed during training, constituting a strict zero-shot resolution extrapolation setting.

This training–testing protocol is designed to evaluate whether a model trained only on limited local visual context can extrapolate to larger, unseen spatial extents. Such capability is essential in computational pathology, where fine-grained annotations are typically available only for small regions, while diagnostic decisions often depend on global tissue organization. Successful performance under this protocol indicates that the learned representations capture scale-invariant and hierarchical tissue structures rather than resolution-specific textures.

**Baseline** We compare DF-Mamba with three representative baseline architectures that cover the major paradigms in visual representation learning: ResNet(He et al., 2016), Vision Transformer (ViT)(Dosovitskiy et al., 2021) with learned absolute 2D positional embeddings and standard bicubic interpolation, and Vision Mamba (ViM) (Zhu et al., 2024). These baselines are selected to evaluate model performance from complementary perspectives, including convolutional inductive bias, attention-based global modeling, and state-space sequence modeling. All baseline models are trained and evaluated under the same resolution extrapolation protocol, using identical data splits, input preprocessing, and training schedules. In particular, all models are trained on local image crops and evaluated on full-resolution images at test time, ensuring that performance differences arise from architectural design rather than training data exposure.

**Result** Table 4 reports the performance and efficiency comparison of DF-Mamba against ResNet, ViT, and ViM under a zero-shot resolution extrapolation setting, where all models are trained on local image crops and evaluated on larger,

*Table 4.* Comparisons of efficiency and accuracy on the Computational Pathology task across different input resolutions ($L_{train} = 64$ and $L_{train} = 256$) as training (evaluated at $L_{test} = 1024$). We benchmark against standard baselines (ViT, ResNet, and ViM)

| METHOD | PARAMS | $L_{train} = 64(S = 128)$ | | | $L_{train} = 256(S = 256)$ | | |
| --- | --- | --- | --- | --- | --- | --- | --- |
| | | FLOPs | INF. (MS) | ACC. (%) | FLOPs | INF. (MS) | ACC. (%) |
| VIT | 2.305M | 77.88M | 220.14 | 57.78 | 1.18G | 240.86 | 57.78 |
| RESNET | 11.18M | 297.72M | **51.82** | 62.22 | 4.76G | 211.87 | 68.89 |
| VIM | 1.920M | 61.26M | 126.15 | 64.44 | 0.98G | **208.34** | 64.44 |
| **DF-MAMBA (OURS)** | **0.89M** | **39.83M** | 68.41 | **71.11** | **0.72G** | 282.19 | **77.78** |
| *w/o Consist. Loss* | 0.89M | 39.83M | 66.40 | 66.67 | 0.72G | 266.26 | 71.11 |
| *w/o Time-Scaler* | 0.89M | 39.83M | 67.10 | 66.67 | 0.72G | 275.83 | 71.11 |

unseen resolutions at test time.

DF-Mamba consistently achieves the best classification accuracy across all resolutions, while using the fewest parameters and maintaining favorable computational complexity. At resolution $L_{train} = 64$, DF-Mamba attains an accuracy of 71.11%, outperforming ResNet (62.22%), ViT (57.78%), and ViM (64.44%). When evaluated at the higher resolution $L_{train} = 256$, DF-Mamba further improves to 77.78%, demonstrating a clear positive scaling behavior under increased spatial context. In contrast, ViT and ViM exhibit performance stagnation across resolutions, with no accuracy gain when the input resolution increases. ResNet shows moderate improvement due to its convolutional inductive bias, but remains significantly inferior to DF-Mamba.

DF-Mamba is highly parameter-efficient, requiring only 0.89M parameters, which is less than half of ViM (1.92M) and ViT (2.30M), and an order of magnitude fewer than ResNet (11.18M). This efficiency stems from the recursive fractal design with shared dynamics across scales, enabling the model to reuse parameters rather than learning scale-specific representations.

In terms of computation, DF-Mamba also achieves the lowest FLOPs at both resolutions (39.83M at $L_{train} = 64$ and 0.72G at $L_{train} = 256$), confirming its linear-time scalability. While ResNet benefits from optimized convolution kernels in terms of latency, its substantially higher FLOPs indicate poor energy efficiency and limited scalability to higher resolutions.

A key observation from Table 4 is the distinct scaling behavior across architectures. DF-Mamba exhibits a clear positive scaling trend: increasing the test resolution substantially improves accuracy, suggesting that DF-Mamba can effectively assimilate additional spatial context to refine global tissue understanding. By contrast, both ViT and ViM fail to benefit from higher-resolution inputs under our zero-shot extrapolation protocol. Notably, this limitation of ViT is not explained by positional-embedding handling: ViT with standard 2D positional-embedding interpolation and ViT with fixed 2D sin-cos embeddings achieve identical accuracy, in-

dicating that the bottleneck lies in cross-scale representation learning. ViM, despite its linear complexity, operates as a flat state-space model without explicit hierarchical coarse-graining, which can cause global structure to be obscured by accumulated local variations. ResNet partially benefits from translation invariance but remains constrained by its fixed effective receptive field, limiting its ability to capture long-range tissue organization. More details are shown in Appendix A.10.5

**Ablation Study** The ablation results in Table 4 further validate the necessity of DF-Mamba's core components. Removing either the consistency loss or the time-scaling mechanism leads to a notable performance degradation at both resolutions. In particular, while the ablated variants still show limited scaling behavior, their accuracy at $L_{train} = 256$ remains more than 6 percentage points below the full model. This confirms that both physical consistency constraints and recursive time-scaling are essential for enforcing scale-consistent representations and enabling robust resolution extrapolation.

Overall, these results demonstrate that DF-Mamba not only achieves state-of-the-art accuracy on computational pathology, but also uniquely satisfies the combined requirements of parameter efficiency, computational scalability, and robust resolution extrapolation. The superior performance under zero-shot scaling provides strong empirical evidence that hierarchical coarse-graining and renormalization-inspired dynamics are critical for modeling multi-scale biological structures.

## 5. Conclusion

We presented DF-Mamba, a scale-aware state-space framework inspired by renormalization group (RG) flow and realized by recursive, parameter-shared coarse-graining. By combining exponential time scaling, information-preserving block-spin downsampling, and RG-gated multi-scale fusion, DF-Mamba maintains overall linear-time complexity in sequence length while capturing dependencies across exponentially separated scales. Across time series forecast-

ing, spatial transcriptomics, and computational pathology, DF-Mamba achieves strong performance and demonstrates zero-shot scale extrapolation in computational pathology, suggesting improved robustness to scale shifts compared to Transformers and standard Mamba baselines. However, DF-Mamba is most beneficial when data exhibits inherent multi-scale structure and tasks require integrating local details with global context; for short or predominantly single-scale signals, gains may be limited.

## Acknowledgements

We thank the reviewers and area chair for their constructive feedback. This work was supported by the Cardiff University Digital Transformation Innovation Institute Seedcorn Funding Scheme. We acknowledge the use of resources provided by the Isambard-AI National AI Research Resource (AIRR). Isambard-AI is operated by the University of Bristol and is funded by the UK Government's Department for Science, Innovation and Technology (DSIT) via UK Research and Innovation; and the Science and Technology Facilities Council [ST/AIRR/I-A-I/1023] (McIntosh-Smith et al., 2024).

## Impact Statement

This paper presents a deep learning framework for scalable and scale-consistent modeling of complex biomedical and scientific data. The proposed architecture is particularly relevant for applications involving multi-scale biological structure, including spatial transcriptomics, computational pathology, medical imaging, and longitudinal clinical data analysis. By enabling efficient integration of local and global patterns across large-scale datasets, this framework may support the development of more robust AI tools for disease stratification, biomarker discovery, and precision medicine. The ability of the model to generalize across unseen temporal and spatial scales while maintaining computational efficiency may be especially valuable for real-world biomedical applications, where training data are often limited and heterogeneous. These properties could facilitate future translational applications in areas such as digital pathology, imaging-based diagnostics, and multimodal clinical decision support systems.

As with all AI methods intended for healthcare and biomedical research, careful validation, transparency, and human oversight remain essential prior to clinical deployment. Potential risks include reduced performance under distributional shift, bias arising from unrepresentative training data, and overreliance on automated predictions in high-stakes settings.

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

# A. appendix

## A.1. Theoretical Framework and RG Isomorphism

In this section, we establish an RG-inspired correspondence between DF-Mamba and the Kadanoff block-spin transformation, and interpret DF-Mamba as a variational point-estimate approximation to the RG flow.

Table 5 establishes the structural correspondence between the physical Renormalization Group (RG) concepts and our specific neural implementations. Crucially, the architectural decisions in DF-Mamba, specifically the learnable block-spin downsampling, the shared recursive dynamics, and the consistency objectives, are not arbitrary engineering heuristics but are principled instantiations grounded in the physical requirements listed in Table 5. For instance, the transition from microscopic degrees of freedom to effective theories motivates our specific choice of recursive weight sharing combined with exponential time-step scaling, ensuring the model adheres to scale-invariant physical constraints, effectively functioning as a variational approximation to the RG flow.

Classical RG coarse-graining can be stochastic and corresponds to integrating out degrees of freedom. DF-Mamba employs a deterministic, learnable downsampler $D_\phi$, which may be viewed as a variational point-estimate (Dirac posterior) approximation to a stochastic coarse-graining transition.

The intended scale separation is encoded by the base dilation law $\bar{\Delta}_k = \bar{\Delta}_0 2^k$. If we additionally use a content-adaptive modulation $m(X)$, we treat it as a bounded engineering refinement and clip $\Delta_k(X) = m(X)\bar{\Delta}_k$ to avoid over-damping at large $k$.

When the continuous dynamics are parameterized with channel-wise (diagonal) negative eigenvalues, e.g., $A = \mathrm{diag}(-\exp(\cdot))$, the ZOH transition $\exp(\Delta_k A)$ has spectral radius $\leq 1$. For more general parameterizations, stability depends on the eigenvalues of $A$.

We use the RG correspondence to motivate architectural and regularization choices (coarse-graining, fusion, and cross-scale constraints), and we do not claim an exact equivalence to RG for arbitrary data distributions.

### A.1.1. PHYSICAL FORMULATION

Consider a 1D statistical physical system defined by a set of microscopic variables (spins) $\mathbf{x} = \{x_1, x_2, \ldots, x_L\}$. The probability distribution of the system is governed by the Boltzmann distribution:

$$P(\mathbf{x}) = \frac{1}{Z} \exp(-\mathcal{H}(\mathbf{x})) \tag{14}$$

where $\mathcal{H}(\mathbf{x})$ is the Hamiltonian (energy function) describing the interactions between spins, and $Z$ is the partition function.

The Real-Space Renormalization Group (RSRG) transformation $\mathcal{R}$ consists of two steps:

1. **Coarse-Graining (Block-Spin):** Grouping adjacent microscopic variables into blocks and defining effective variables $\mathbf{h}'$.

2. **Rescaling:** Rescaling the spatial/temporal dimensions to restore the original density.

Mathematically, the coarse-graining step is an integration over microscopic degrees of freedom while keeping the macroscopic variables fixed:

$$P'(\mathbf{h}') = \int \mathcal{T}(\mathbf{h}'|\mathbf{x})P(\mathbf{x})d\mathbf{x} \tag{15}$$

where $\mathcal{T}(\mathbf{h}'|\mathbf{x})$ is the projection operator (e.g., majority rule or decimation). The goal of RG is to find the effective Hamiltonian $\mathcal{H}'$ such that $P'(\mathbf{h}') \propto \exp(-\mathcal{H}'(\mathbf{h}'))$.

## A.2. Neural Implementation: A Variational Perspective

Unlike classical statistical mechanics where spins fluctuate thermally, standard deep neural networks operate deterministically. To bridge this gap, we formulate DF-Mamba as a **variational point-estimate approximation** to the RG flow.

**1. The Induced Distribution $P_k(\mathbf{h})$.** Let $P_{\text{data}}(\mathbf{x})$ be the empirical distribution of the input data. The latent representation $\mathbf{H}^{(k)}$ at depth $k$ is a random variable whose distribution $P_k(\mathbf{h})$ is the *push-forward measure* induced by the network's

*Table 5.* Math/Physics/Model correspondence: fractal geometry notions and block-spin RG concepts mapped to DF-Mamba components and objectives.

| Fractal geometry (math) | RG concept (physics) | DF-Mamba instantiation (DL) |
|---|---|---|
| Microscopic structure across resolutions; fine-scale DoF underpin macroscopic patterns | Microscopic degrees of freedom (DoF) at scale $k$: $\{s_i\}$ or state field $h^{(k)}$ | Token-level hidden states at depth/scale $k$: $H^{(k)} \in \mathbb{R}^{L_k \times D}$ ("local view") |
| Hierarchical self-similarity emerges through repeated coarse descriptions | Block-spin coarse-graining operator $T$: $h^{(k)} \mapsto h^{(k+1)}$ (often stochastic / marginalization) | Learnable downsampling (pair-wise coarse-graining): $X_{k+1} = D_\phi(P_k)$ where $P_k = \text{PairConcat}(H_{\text{local}}^{(k)})$ |
| Multi-scale dynamics: coarse scales evolve more slowly (long-range dependencies) | Effective dynamics at coarser scale: RG flow induces effective parameters / couplings | Shared SSM/Mamba dynamics at each scale: $H_{\text{local}}^{(k)} = \Phi_\theta(X_k, k)$ with scale-aware base step $\bar{\Delta}_k = \bar{\Delta}_0 2^k$ (optionally modulated by bounded $m(X)$) |
| From integrating out fine DoF to an effective theory retaining relevant statistics | Integrating out DoF yields an effective theory (effective description at coarse scale) | (Operationalized by) coarse representation $X_{k+1}$ produced via $D_\phi(\cdot)$; optionally regularized by reconstruction to retain information |
| Statistical self-similarity / scale invariance (fixed-point behavior) | Fixed points represent self-similarity (invariance under coarse-graining) | Scale-consistency regularizer: $L_{\text{consist}} = \sum_k \|H_{\text{global}}^{(k)} - \text{sg}(H_{\text{local}}^{(k)})\|_2^2$ |
| Coupling fine and coarse descriptions (multi-resolution synthesis) | Cross-scale fusion (micro/macro coupling across scales) | Cross-scale fusion: $H_{\text{out}}^{(k)} = (1 - g) \odot H_{\text{local}}^{(k)} + g \odot H_{\text{global}}^{(k)}, g = \sigma\big(G([H_{\text{local}}^{(k)}, H_{\text{global}}^{(k)}])\big)$ |
| (Analysis tool) approximate inverse mapping to inspect information retained across scales | Inverse mapping / decoding from coarse to fine (not always defined in RG; used for analysis) | Learnable upsampling / reconstruction surrogate: $\hat{P}_k = D_\psi^\dagger(X_{k+1})$, used only for auxiliary objectives |
| Information preservation across scales (no catastrophic loss of relevant content) | Information preservation across scales (idealized) | Reconstruction regularizer: $L_{\text{recon}} = \sum_k \|\text{sg}(P_k) - D_\psi^\dagger(D_\phi(P_k))\|_2^2$ |
| Smooth evolution across scales (avoid abrupt, unstable scale transitions) | Smooth RG flow (parameters change gradually with scale) | Gate smoothness (optional): $L_{\text{flow}} = \sum_k \|\bar{g}_k - \text{sg}(\bar{g}_{k+1})\|_2^2, \bar{g}_k = \text{mean}(g^{(k)})$ |

deterministic mapping $\Phi_{0 \to k}$ up to that layer:

$$P_k(\mathbf{h}) = \int \delta(\mathbf{h} - \Phi_{0 \to k}(\mathbf{x})) P_{\text{data}}(\mathbf{x}) d\mathbf{x} \qquad (16)$$

where $\delta(\cdot)$ is the Dirac delta function.

**2. The Variational Approximation.** The exact RG transformation requires integrating over microscopic fluctuations to obtain the marginal $P_{k+1}(\mathbf{h}')$. This is computationally intractable. We approximate the stochastic coarse-graining operator $\mathcal{T}(\mathbf{h}'|\mathbf{h})$ with a deterministic parameterized function $\mathcal{D}_\phi(\mathbf{h})$. In variational inference terms, this corresponds to assuming a **Dirac variational posterior**:

$$q_\phi(\mathbf{h}'|\mathbf{h}) = \delta(\mathbf{h}' - \mathcal{D}_\phi(\mathbf{h})) \qquad (17)$$

Under this assumption, the intractable integral collapses to a function evaluation:

$$\mathbf{H}^{(k+1)} = \int \mathbf{h}' \cdot q_\phi(\mathbf{h}'|\mathbf{H}^{(k)}) d\mathbf{h}' = \mathcal{D}_\phi(\mathbf{H}^{(k)}) \qquad (18)$$

**3. Information-Theoretic Justification.** $\mathcal{L}_{recon}$ We justify MSE reconstruction as maximizing a variational lower bound on mutual information across scales (Barber–Agakov). Let $\mathbf{P}_k$ denote the fine-scale blocked representation and $\mathbf{X}_{k+1}$ the coarse-scale representation. Mutual information satisfies

$$I(\mathbf{P}_k; \mathbf{X}_{k+1}) = H(\mathbf{P}_k) - H(\mathbf{P}_k|\mathbf{X}_{k+1}). \qquad (19)$$

For any variational conditional density $q_\theta(\mathbf{P}_k|\mathbf{X}_{k+1})$,

$$H(\mathbf{P}_k|\mathbf{X}_{k+1}) \leq -\mathbb{E}\big[\log q_\theta(\mathbf{P}_k|\mathbf{X}_{k+1})\big], \qquad (20)$$

which yields the lower bound

$$I(\mathbf{P}_k; \mathbf{X}_{k+1}) \geq H(\mathbf{P}_k) + \mathbb{E}\big[\log q_\theta(\mathbf{P}_k|\mathbf{X}_{k+1})\big]. \tag{21}$$

We instantiate $q_\theta$ as the decoder $\mathcal{D}^\dagger$ and assume a Gaussian likelihood with fixed variance $\sigma^2$:

$$q_\theta(\mathbf{P}_k|\mathbf{X}_{k+1}) = \mathcal{N}\big(\mathcal{D}^\dagger(\mathbf{X}_{k+1}),\, \sigma^2\mathbf{I}\big), \qquad \log q_\theta(\mathbf{P}_k|\mathbf{X}_{k+1}) = \text{const} - \frac{\|\mathbf{P}_k - \mathcal{D}^\dagger(\mathbf{X}_{k+1})\|_2^2}{2\sigma^2}. \tag{22}$$

Therefore, maximizing $\mathbb{E}[\log q_\theta]$ is equivalent (up to an additive constant and scaling) to minimizing an MSE reconstruction loss.

Importantly, Eq. 11 applies a stop-gradient to the reconstruction target $\text{sg}(\mathbf{P}_k)$, so during the update of the downsampler/decoder parameters $\theta$, the source distribution of $\mathbf{P}_k$ is treated as fixed. Under this condition, the entropy term $H(\mathbf{P}_k)$ in Eq. (21) is constant w.r.t. $\theta$ for the local optimization step. Consequently, minimizing $\mathcal{L}_{recon}$ is *locally* equivalent to maximizing the variational lower bound on $I(\mathbf{P}_k; \mathbf{X}_{k+1})$ with respect to $\theta$.

### A.3. Continuous Dynamics and Time Dilation

Let the continuous dynamics of the latent state $h(t)$ be governed by the linear ODE:

$$\frac{dh(t)}{dt} = \mathbf{A}h(t) + \mathbf{B}x(t) \tag{23}$$

where $t$ represents the microscopic time unit.

At recursion depth $k$, the sequence length is reduced by a factor of $2^k$. This implies a geometric **Time Dilation**. Let $\tau$ be the effective time variable at depth $k$. The relationship between microscopic time $t$ and effective time $\tau$ is:

$$t = 2^k \cdot \tau \implies dt = 2^k d\tau \tag{24}$$

Substituting this into the ODE:

$$\frac{dh}{2^k d\tau} = \mathbf{A}h + \mathbf{B}x \implies \frac{dh}{d\tau} = (2^k\mathbf{A})h + (2^k\mathbf{B})x \tag{25}$$

#### A.3.1. DISCRETIZATION CONSISTENCY

The State Space Model discretizes the continuous system using the Zero-Order Hold (ZOH) rule. For a step size $\Delta$, the discrete transition matrix $\overline{\mathbf{A}}$ is $\exp(\Delta\mathbf{A})$.

To maintain **Scale Invariance**, the discrete transition learned by the shared SSM core at level $k$ must be equivalent to evolving the original system for an interval of $2^k\Delta_0$. Let $\Delta_k$ be the discretization step applied at level $k$. The discrete evolution is:

$$\overline{\mathbf{A}}_k = \exp(\Delta_k\mathbf{A}) \tag{26}$$

From the physical requirement derived in Eq. (25), the effective $\mathbf{A}$ matrix scales by $2^k$. Alternatively, viewing this from the time domain:

$$\exp(\Delta_k\mathbf{A}) \equiv \exp(2^k\Delta_0\mathbf{A}) \tag{27}$$

Comparing the exponents yields the scaling law:

$$\Delta_k = \Delta_0 \cdot 2^k \tag{28}$$

This proves that the discretization step $\Delta$ must scale exponentially with depth $k$ to preserve the physical consistency of the Mamba.

It is crucial to emphasize that the scaling rule $\Delta_k = \Delta_0 \cdot 2^k$ is not an ad-hoc design choice. Instead, it is a mathematical consequence of enforcing *discretization consistency* under *shared continuous-time dynamics* $(A, B)$ across recursion depths. Concretely, interpreting deeper recursion as operating on a coarser scale with dilated physical time ($dt = 2^k d\tau$) implies that maintaining the same underlying continuous system requires the discrete step to scale as $\Delta_k \propto 2^k$.

Standard multi-scale backbones (e.g., U-Net-style encoder–decoders or multi-stage pyramids) typically enlarge the receptive field via pooling or striding, but treat the internal state processing as scale-agnostic, lacking an explicit constraint that

couples the scale level to the continuous-time discretization. In contrast, DF-Mamba *explicitly ties* the scale index $k$ to the SSM discretization via $\Delta_k = \Delta_0 2^k$, thereby injecting a principled inductive bias toward scale-consistent dynamics. We frame this as an RG-inspired structural constraint, rather than a claim of exact equivalence to physical RG, and empirically find it crucial for robust cross-scale generalization.

### A.4. Gradient Iso-Normality and Spectral Regularization

Training deep recursive networks is prone to vanishing or exploding gradients, which are governed by the spectrum of Jacobian products along depth. In this section, we provide a principled analysis showing that the proposed **Consistency Loss** acts as a *spectral regularizer*, encouraging an *approximate, local form of dynamical isometry* along the data manifold.

### A.5. Jacobian Dynamics in Recursive Flows

Let $\mathcal{F}_\theta$ denote the recursive mapping that transforms a local representation $\mathbf{h}_k$ into its corresponding global context $\mathbf{h}_k^{\text{global}}$. The backward gradient propagation across recursion depth depends on the Jacobian

$$\mathbf{J}_{\mathcal{F}}^{(k)} = \frac{\partial \mathbf{h}_k^{\text{global}}}{\partial \mathbf{h}_k}. \tag{29}$$

For a recursion of depth $K$, the gradient norm satisfies the standard bound

$$\|\nabla_{\mathbf{x}}\mathcal{L}\| \leq \|\nabla_{\mathbf{y}}\mathcal{L}\| \cdot \prod_{k=0}^{K-1} \|\mathbf{J}_{\mathcal{F}}^{(k)}\|_{op}, \tag{30}$$

where $\|\cdot\|_{op}$ denotes the operator (spectral) norm. Stable optimization requires $\|\mathbf{J}_{\mathcal{F}}^{(k)}\|_{op}$ to remain close to 1 along the recursion; otherwise, gradients may vanish or explode exponentially.

### A.6. Consistency Loss as Identity-Regularizing Objective

The Consistency Loss is defined using a stop-gradient operator $\text{sg}(\cdot)$:

$$\mathcal{L}_{\text{consist}} = \mathbb{E}\left[\left\|\mathcal{F}_\theta(\mathbf{h}_k) - \text{sg}(\mathbf{h}_k)\right\|_2^2\right]. \tag{31}$$

The stop-gradient operation is crucial: it ensures that optimization updates only the parameters $\theta$ of $\mathcal{F}_\theta$, while treating $\mathbf{h}_k$ as a fixed anchor. This prevents trivial collapse (e.g., $\mathbf{h}_k \to \mathbf{0}$) and decouples the optimization of the mapping from the evolution of the representation itself.

Minimizing $\mathcal{L}_{\text{consist}}$ encourages $\mathcal{F}_\theta$ to behave as an approximate identity mapping on the data manifold $\mathcal{M}$:

$$\mathcal{F}_\theta(\mathbf{h}) \approx \mathbf{h}, \quad \forall \mathbf{h} \in \mathcal{M}. \tag{32}$$

Importantly, this constraint is enforced only locally along $\mathcal{M}$, rather than globally over the entire input space.

### A.7. Local Spectral Bound via Residual Decomposition

Assume that $\mathcal{F}_\theta$ is $L$-smooth in a neighborhood of $\mathcal{M}$. When $\mathcal{L}_{\text{consist}}$ is sufficiently small, we can express the mapping as

$$\mathcal{F}_\theta(\mathbf{h}) = \mathbf{h} + \mathbf{r}(\mathbf{h}), \tag{33}$$

where the residual satisfies $\|\mathbf{r}(\mathbf{h})\|_2 \leq \varepsilon$ for $\mathbf{h} \in \mathcal{M}$.

The Jacobian then decomposes as

$$\mathbf{J}_{\mathcal{F}}(\mathbf{h}) = \mathbf{I} + \nabla \mathbf{r}(\mathbf{h}). \tag{34}$$

Under the smoothness assumption, the magnitude of $\nabla \mathbf{r}(\mathbf{h})$ is empirically constrained during training. Applying Weyl's inequality to the singular values yields the local bound

$$1 - \|\nabla \mathbf{r}(\mathbf{h})\|_{op} \ \leq \ \rho(\mathbf{J}_{\mathcal{F}}(\mathbf{h})) \ \leq \ 1 + \|\nabla \mathbf{r}(\mathbf{h})\|_{op}, \tag{35}$$

where $\rho(\cdot)$ denotes the spectral radius.

Consequently, minimizing $\mathcal{L}_{\text{consist}}$ encourages the Jacobian spectrum to remain concentrated near unity *along the data manifold*, yielding an **approximate, local form of dynamical isometry**. This spectral regularization mitigates gradient distortion across recursive depth and empirically stabilizes training of deep fractal architectures.

**Remark.**   We emphasize that this analysis characterizes local Jacobian behavior induced by the consistency objective and does not guarantee global spectral bounds. Nevertheless, it provides a principled explanation for the observed stability of DF-Mamba under deep recursive composition.

## A.8. Complexity and Memory Analysis

We formally prove that DF-Mamba maintains linear computational complexity $O(L)$, contrasting it with the quadratic complexity $O(L^2)$ of Transformers.

**Theorem 1.** *The total computational cost of DF-Mamba for a sequence of length $L$ is linear with respect to $L$.*

*Proof.* Let $\mathcal{C}_{\text{Mamba}}(N)$ denote the floating-point operations (FLOPs) required to process a sequence of length $N$ using a standard Mamba block. A key property of the Selective State Space Model, particularly with the hardware-aware parallel scan implementation, is that its complexity scales linearly with the sequence length. We can express this as:

$$\mathcal{C}_{\text{Mamba}}(N) = \kappa(D, d_{state}) \cdot N \tag{36}$$

where $\kappa(D, d_{state})$ is a constant factor depending on the model width $D$, expansion factor $E$, and state dimension $d_{state}$, but independent of $N$.

In the DF-Mamba architecture, the sequence length is halved at each recursive depth $k$. At depth $k$, the effective sequence length is $L_k = L/2^k$. The total computational cost is the sum of costs across all recursive levels $k = 0, 1, \ldots, K$ (where $K \approx \log_2 L$):

$$\mathcal{C}_{total} = \sum_{k=0}^{K} \mathcal{C}_{\text{Mamba}}\left(\frac{L}{2^k}\right) = \sum_{k=0}^{K} \kappa \cdot \frac{L}{2^k} \tag{37}$$

Factoring out the constant terms $\kappa \cdot L$, we obtain a geometric series:

$$\mathcal{C}_{total} = \kappa \cdot L \cdot \left(1 + \frac{1}{2} + \frac{1}{4} + \cdots + \frac{1}{2^K}\right) \tag{38}$$

As $K \to \infty$, the geometric series $\sum_{k=0}^{\infty}(1/2)^k$ converges to 2. Therefore:

$$\mathcal{C}_{total} < 2 \cdot \kappa \cdot L = 2 \cdot \mathcal{C}_{\text{Mamba}}(L) \tag{39}$$

Consequently:

$$\mathcal{C}_{total} = O(L) \tag{40}$$

This derivation demonstrates that DF-Mamba incurs only a constant bounded overhead (approximately $2\times$) compared to a single-scale non-hierarchical Mamba, while effectively expanding the receptive field to the global context.    $\square$

**Memory Complexity.**   In practice, DF-Mamba can be implemented with linear memory complexity. At inference time, only the current-scale states need to be retained, yielding $O(L)$ memory usage. During training, intermediate activations across recursion depths can be recomputed or checkpointed, resulting in $O(L)$ memory up to constant factors.

## A.9. Algorithm Details

For clarity and reproducibility, we provide the detailed pseudocode for a single layer of our DF-Mamba mechanism in Algorithm 1.

## A.10. More Details of Results

### A.10.1. COMMON EXPERIMENTAL SETTINGS.

We implement DF-Mamba using PyTorch and conduct all experiments on a cluster equipped with one NVIDIA GH200 GPU. The model is optimized using the AdamW optimizer with an initial learning rate of $1 \times 10^{-4}$, a weight decay of 0.05, and gradient clipping with a max norm of 1.0 to ensure stability. We employ a cosine annealing scheduler with a linear warm-up for the first 10% of epochs. Unless otherwise stated, training runs for 200 epochs with a global batch size of 64, with early stopping enabled (patience = 10 epochs). To ensure reproducibility, a fixed random seed of 42 is applied across all experimental runs. Performance is evaluated using standard metrics specific to each domain, as detailed in the subsequent subsections.

### A.10.2. MORE DETAILS OF DF-MAMBA-TS

Figure 2 visualizes the *Fractal Gate* in DF-Mamba on ETT benchmarks using a long look-back window ($L = 336$). The gate controls the interpolation between local features and global representations, where the colorbar denotes *Gate* ($0 =$ Local, $1 =$ Global). Overall, both datasets exhibit high gate activations across most variables and time steps, indicating that DF-Mamba relies strongly on coarse-grained context when processing long histories. Moreover, ETTm2 (Fig. 2b) shows gate values that are closer to saturation near the global end, suggesting an even stronger preference for coarse/global representations in minute-level forecasting, which is consistent with the higher-frequency fluctuations typical of minute-scale measurements. In contrast, ETTh2 (Fig. 3b) remains strongly global but presents comparatively larger temporal/variable-specific variations, reflecting a more balanced and input-adaptive trade-off between local and global information.

Figure 3 compares the Fractal Gate activations on ETTh2 when using a short look-back window ($L = 96$) versus a long look-back window ($L = 336$). The gate controls the interpolation between local and global representations, where *Gate*$= 0$ indicates purely local reliance and *Gate*$= 1$ indicates purely global reliance. We observe a systematic shift toward larger gate values when increasing the available history from $L = 96$ to $L = 336$, indicating that DF-Mamba relies more on coarse/global representations when longer context is provided. This behavior is consistent with the model's RG-style recursive computation: longer look-back windows make coarse-grained context more informative for stabilizing long-range dynamics, while shorter windows retain a comparatively stronger dependence on local information.

To complement the gate/saliency analyses and mitigate concerns of cherry-picking, we visualize qualitative forecasts on the target variable OT using randomly sampled instances from the test split.

Figure 4 reports multiple random cases for two ETT benchmarks (ETTm2 and ETTh2), where the gray curve denotes the look-back window ($L = 336$), the green curve denotes the ground-truth future over the prediction horizon ($H = 96$), and the red dashed curve is the model prediction (all shown in normalized scale).

Across random samples, DF-Mamba generally preserves the global trend and phase of the future trajectory. The main failure modes occur around sharp turning points and peak regions, where the predictions may exhibit mild temporal lag and/or amplitude attenuation, which is more noticeable in sequences with stronger high-frequency oscillations.

Overall, the random-case visualizations support that the model's performance is not driven by a single selected example and is consistent with the multi-scale behavior observed in the gate analysis. While minor deviations occur at sharp turning points or high-frequency peaks (manifesting as mild lag or amplitude attenuation), this behavior is consistent with the physical principles of renormalization: the model prioritizes **dominant macroscopic dynamics** over high-frequency local fluctuations, effectively acting as a learnable physics-informed filter against noise.

To verify that DF-Mamba learns scale-invariant representations regardless of input resolution, we evaluate its performance on ETTh2 and ETTm2 with varying look-back windows $L \in \{96, 192, 336\}$. As shown in Table 6, the model exhibits *consistent stability* across different context lengths. While shorter windows ($L = 96$ or $192$) occasionally yield marginally lower errors by avoiding high-frequency noise accumulation in these specific datasets, the performance gap between these and the longer context ($L = 336$) is minimal. This observation aligns with our design goal: the recursive coarse-graining mechanism effectively filters redundancy, allowing the model to remain robust even when fed with extended histories. These results confirm that DF-Mamba does not strictly rely on a carefully tuned window size, justifying our adoption of the standardized $L = 336$ setting in the main experiments for fair comparison with baselines.

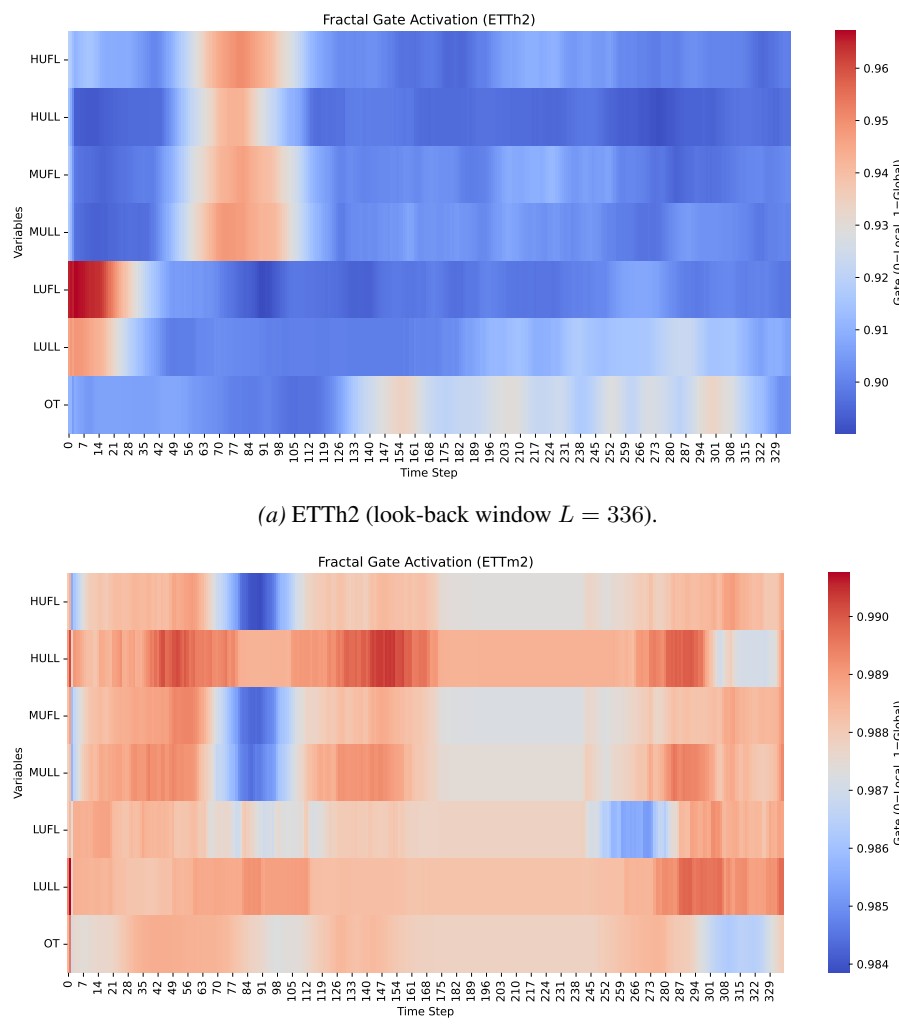

*(a)* ETTh2 (look-back window $L = 336$).

*(b)* ETTm2 (look-back window $L = 336$).

*Figure 2.* **Fractal gate activation over long look-back windows.** Gate values control the fusion between local (fine-scale) and global (coarse-scale) representations in DF-Mamba, where the colorbar denotes *Gate* ($0$ = Local, $1$ = Global). The x-axis spans the historical look-back window ($L = 336$) and the y-axis lists variables.

### A.10.3. MORE DETAILS OF DF-MAMBA-ST

In the main paper, we primarily report ARI and NMI, which are widely used to measure the overall agreement between predicted clusters and ground-truth annotations. To further characterize the structural failure mode that is particularly relevant in spatial transcriptomics, we additionally report Completeness. Completeness evaluates to what extent samples sharing the same ground-truth label are assigned to the same predicted cluster. This perspective is complementary to ARI/NMI, while ARI and NMI summarize global partition agreement, Completeness specifically emphasizes the ability to keep each biological domain cohesive rather than scattered.

As shown in Table 7, DF-Mamba-ST achieves an average Completeness of 0.64 on DLPFC, ranking second overall and remaining close to the strongest baseline DiffusionST (0.65), while outperforming BayesSpace, SpaGCN, DeepST, GraphST, and BASS in aggregate. Notably, on the Breast Cancer dataset, DF-Mamba-ST obtains the best Completeness score (0.69), surpassing the strongest baseline GraphST (0.64) by a clear margin.

These results suggest that DF-Mamba-ST is particularly effective at preventing domain fragmentation, producing more cohesive and interpretable tissue partitions. We attribute this advantage to the recursive, multi-scale modeling in DF-Mamba (RG-inspired coarse-graining and scale-aware fusion), which encourages consistent region-level representations across

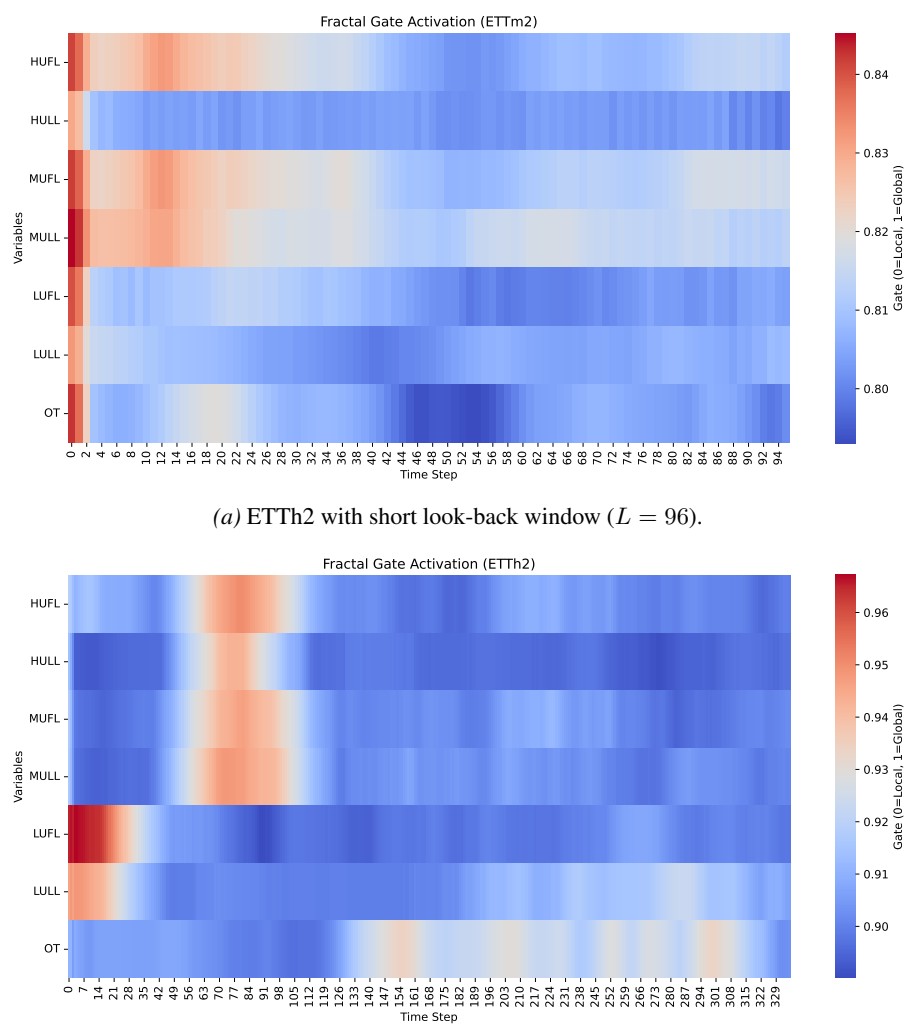

*(a)* ETTh2 with short look-back window ($L = 96$).

*(b)* ETTh2 with long look-back window ($L = 336$).

*Figure 3.* **Effect of look-back window length on fractal gate activations (ETTh2).** DF-Mamba uses a Fractal Gate to fuse local (fine-scale) and global (coarse-scale) representations; the colorbar denotes *Gate* ($0 = $ Local, $1 = $ Global). The x-axis spans the historical look-back window and the y-axis lists variables. *Note:* the two panels may use different colorbar ranges; thus, absolute comparisons should be made based on the numeric values rather than color intensity.

spatial resolutions, thereby stabilizing cluster assignment for each biological domain.

### A.10.4. MORE DETAILS OF COMPUTATIONAL PATHOLOGY

To validate that DF-Mamba captures scale-consistent physical features rather than relying on spurious correlations, we employ SmoothGrad to visualize the saliency maps of the same pathological tissue samples at two distinct resolutions: coarse-grained ($L_{train} = 64$) and fine-grained ($L_{train} = 256$).

Figures 5a and 5b display the original inputs, saliency maps, and overlays for four representative cases: Normal, InSitu, Benign, and Invasive. A comparative analysis reveals the following insights regarding the model's internal representations:

Comparing the heatmap overlays between $L_{train} = 64$ and $L_{train} = 256$, we observe a strong spatial alignment in the regions of interest. For example, in the Benign case, both resolutions highlight the epithelial lining of the glandular structures. This suggests that DF-Mamba attends to scale-consistent morphological cues that persist across magnifications, supporting the effectiveness of the RG-inspired recursive architecture.

he visual difference in saliency maps is consistent with an RG-style coarse-to-fine interpretation:

*Table 6.* Sensitivity to look-back window $L$ on ETTh2 and ETTm2. We report MSE and MAE under $L \in \{96, 192, 336\}$ for different prediction horizons.

| LOOK-BACK WINDOW | | $L = 96$ | | $L = 192$ | | $L = 336$ | |
|---|---|---|---|---|---|---|---|
| DATASET | PRED | MSE | MAE | MSE | MAE | MSE | MAE |
| ETTH2 | 96 | 0.238 | 0.322 | 0.222 | 0.314 | 0.232 | 0.323 |
| | 192 | 0.287 | 0.361 | 0.284 | 0.361 | 0.287 | 0.364 |
| | 336 | 0.318 | 0.377 | 0.344 | 0.404 | 0.329 | 0.393 |
| | 720 | 0.390 | 0.424 | 0.414 | 0.443 | 0.408 | 0.437 |
| ETTM2 | 96 | 0.157 | 0.257 | 0.155 | 0.258 | 0.158 | 0.261 |
| | 192 | 0.196 | 0.296 | 0.192 | 0.290 | 0.193 | 0.293 |
| | 336 | 0.242 | 0.327 | 0.240 | 0.325 | 0.239 | 0.328 |
| | 720 | 0.302 | 0.366 | 0.299 | 0.362 | 0.316 | 0.375 |

*Table 7.* Clustering completeness on spatial transcriptomics benchmarks. Higher is better.

| DATASET | SLICE | BAYESSPACE | SPAGCN | DEEPST | GRAPHST | BASS | DIFFUSIONST | DF-MAMBA-ST |
|---|---|---|---|---|---|---|---|---|
| DLPFC | 151507 | 0.61 | 0.59 | 0.64 | 0.67 | 0.63 | 0.62 | 0.63 |
| | 151508 | 0.59 | 0.56 | 0.59 | 0.56 | 0.61 | 0.63 | 0.62 |
| | 151509 | 0.61 | 0.63 | 0.64 | 0.57 | 0.63 | 0.66 | 0.65 |
| | 151510 | 0.55 | 0.57 | 0.62 | 0.58 | 0.56 | 0.59 | 0.62 |
| | 151669 | 0.65 | 0.55 | 0.63 | 0.45 | 0.51 | 0.64 | 0.59 |
| | 151670 | 0.60 | 0.54 | 0.53 | 0.40 | 0.46 | 0.63 | 0.57 |
| | 151671 | 0.64 | 0.68 | 0.69 | 0.63 | 0.63 | 0.71 | 0.67 |
| | 151672 | 0.53 | 0.63 | 0.60 | 0.66 | 0.63 | 0.64 | 0.66 |
| | 151673 | 0.70 | 0.62 | 0.69 | 0.65 | 0.64 | 0.73 | 0.68 |
| | 151674 | 0.48 | 0.53 | 0.65 | 0.60 | 0.69 | 0.71 | 0.68 |
| | 151675 | 0.63 | 0.55 | 0.66 | 0.65 | 0.69 | 0.58 | 0.61 |
| | 151676 | 0.54 | 0.48 | 0.63 | 0.58 | 0.68 | 0.69 | 0.67 |
| | AVERAGE | 0.59 | 0.58 | 0.63 | 0.58 | 0.61 | **0.65** | 0.64 |
| BREAST CANCER | – | 0.52 | 0.55 | 0.57 | 0.64 | 0.59 | 0.61 | **0.69** |

- **At $L_{train} = 64$ (Macroscopic View):** The saliency maps are diffuse and contiguous. The model focuses on the global tissue architecture, such as the overall arrangement of glandular lobules and the continuity of tissue compartments. This corresponds to the coarse-grained state where high-frequency texture noise has been integrated out, leaving only the relevant macroscopic operators.

- **At $L_{train} = 256$ (Microscopic View):** The saliency maps become sparse and precise. The model attention shifts to high-frequency details, such as nuclear atypia, chromatin texture, and discrete boundaries between epithelium and stroma.

In the Normal case, the prediction confidence increases significantly from $L_{train} = 64$ (logit $= 1.39$) to $L_{train} = 256$ (logit $= 3.25$). The visualization shows that at $L_{train} = 256$, the model strongly attends to the uniform distribution of healthy nuclei, a feature that is blurred at $L_{train} = 64$. This demonstrates that DF-Mamba effectively utilizes the additional information present at finer scales to bolster its decision-making, while still maintaining correctness at coarser scales.

Interestingly, the InSitu sample is misclassified as Normal at both resolutions (though with higher confidence at $L = 256$). The saliency maps reveal that the model focuses on the gland interior. Carcinoma In Situ is characterized by neoplastic cells contained within the basement membrane. The misclassification suggests that for this specific difficult sample, the subtle morphological cues distinguishing InSitu from Normal (such as slight nuclear crowding) might require even higher resolution or broader contextual integration. However, the consistent misclassification across scales indicates that the model relies on a similar set of cues at both resolutions; this suggests cross-scale consistency of its evidence patterns, although the sample remains challenging.

**Positional-Embedding Fairness for ViT under Zero-shot Scale Extrapolation**  A common concern in zero-shot scale extrapolation is whether ViT-style baselines are disadvantaged by positional embeddings when the token grid increases at test time. In particular, learned absolute positional embeddings are tied to a fixed token length and require resizing to handle

*Table 8.* Comparisons of efficiency and zero-shot scale-extrapolation accuracy on the Computational Pathology task. Models are trained on local crops of size $L_{train} = 64$ or $L_{train} = 256$ and evaluated on larger fields-of-view without fine-tuning. We benchmark against standard baselines (ViT, ResNet), vision Mamba (ViM) **and strong hierarchical architectures (Swin Transformer, ResNeSt, PVTv2)**. For ViT, we report positional-encoding variants: ViT-Interp uses learned absolute PE with 2D bicubic interpolation; ViT-SinCos uses fixed 2D sin-cos PE; ViT-NoInterp uses learned absolute PE without resizing and fails under token-grid change.

| METHOD | PARAMS | $L_{train} = 64 (S = 128)$ | | | $L_{train} = 256 (S = 256)$ | | |
|---|---|---|---|---|---|---|---|
| | | FLOPs | INF. (MS) | ACC. (%) | FLOPs | INF. (MS) | ACC. (%) |
| VIT-INTERP | 2.305M | 77.88M | 220.14 | 57.78 | 1.18G | 240.86 | 57.78 |
| VIT-SINCOS | 2.305M | 77.88M | 264.47 | 57.78 | 1.18G | 295.82 | 57.78 |
| VIT-NOINTERP | 2.305M | 77.88M | 233.90 | N/A[†] | 1.18G | 252.18 | N/A[†] |
| RESNET | 11.18M | 297.72M | **51.82** | 62.22 | 4.76G | 211.87 | 68.89 |
| VIM | 1.920M | 61.26M | 126.15 | 64.44 | 0.98G | **208.34** | 64.44 |
| SWIN-TRANSFORMER | 27.499M | 819.30M | 519.88 | 62.22 | 14.22G | 995.87 | 64.44 |
| RESNEST | 25.442M | 889.56M | 614.14 | 64.44 | 14.18G | 1093.61 | 68.89 |
| PVTV2 | 3.406M | 86.30M | 360.05 | 62.22 | 1.38G | 537.98 | 71.11 |
| **DF-MAMBA (OURS)** | **0.89M** | **39.83M** | 68.41 | **71.11** | **0.72G** | 282.19 | **77.78** |
| *w/o Consist. Loss* | 0.89M | 39.83M | 66.40 | 66.67 | 0.72G | 266.26 | 71.11 |
| *w/o Time-Scaler* | 0.89M | 39.83M | 67.10 | 66.67 | 0.72G | 275.83 | 71.11 |

[†]ViT-NoInterp fails under zero-shot scale extrapolation due to shape mismatch of fixed learned absolute positional embeddings when the token grid increases (no resizing/interpolation).

larger spatial grids. To ensure a fair comparison, we evaluate ViT baselines under standard positional-encoding choices that remain well-defined when the test resolution exceeds the training resolution.

We follow the same zero-shot extrapolation setting as in the main paper.

We report three ViT variants:

(i) **ViT-Interp (Main paper)**: learned absolute 2D positional embeddings resized to the test grid via standard bicubic interpolation;

(ii) **ViT-SinCos**: fixed 2D sin-cos positional embeddings (no resizing needed);

(iii) **ViT-NoInterp**: learned absolute positional embeddings without resizing.

The last variant cannot be evaluated when $S_{\text{test}} > S_{\text{train}}$ because the positional embedding shape mismatches the longer token sequence, and is therefore marked as N/A (Fail).

Table 8 reports accuracy together with efficiency metrics. Notably, **ViT-Interp and ViT-SinCos achieve identical accuracy** under our protocol, indicating that the observed performance gap to DF-Mamba is not an artifact of positional embedding resizing. DF-Mamba consistently achieves higher extrapolation accuracy while using fewer parameters than ViT-style baselines.

A.10.5. DETAILS OF ADDITIONAL HIERARCHICAL BASELINES

To strictly validate the effectiveness of DF-Mamba's recursive coarse-graining mechanism, we extend our comparison beyond standard flat architectures. to include three widely recognized strong multi-scale architectures. These baselines are selected because they explicitly incorporate structural hierarchies to model features at different resolutions, providing a more rigorous test for our proposed zero-shot scale extrapolation capability.

- **Swin Transformer** (Liu et al., 2021) is a representative hierarchical Vision Transformer. It constructs a hierarchical feature representation by starting with small patches and progressively merging neighboring tokens in deeper layers. Its key innovation, the shifted window attention mechanism, allows for efficient local modeling while establishing cross-window connections. We include Swin Transformer to test whether DF-Mamba's recursive dynamic is superior to standard structural hierarchies in handling unseen global scales.

*Table 9.* Zero-shot local-to-global scale extrapolation on NWPU-RESISC45. Models are trained on local crops ($L_{train} = 64$ and $L_{train} = 256$, corresponding to crop sizes $S = 128$ and $S = 256$) and evaluated on full-resolution images ($L_{test} = 1024$) without fine-tuning. We use a stratified split with 10% training, 45% validation, and 45% testing.

| METHOD | $L_{train} = 64 \, (S = 128)$ | $L_{train} = 256 \, (S = 256)$ |
|---|---|---|
| VIT | 18.08% | 47.78% |
| RESNET | 19.74% | 67.96% |
| VIM | 22.53% | 50.13% |
| SWIN TRANSFORMER | 27.33% | 52.57% |
| PVTV2 | 25.62% | 51.25% |
| RESNEST | 23.10% | 61.28% |
| **DF-MAMBA (OURS)** | **43.73%** | **70.30%** |
| *w/o Consist. Loss* | 37.87% | 67.52% |
| *w/o Time-Scaler* | 33.48% | 65.37% |

- **PVTv2** (Wang et al., 2022) (Pyramid Vision Transformer v2) improves upon the original ViT by introducing a progressive shrinking pyramid structure, similar to traditional CNNs. By reducing the sequence length of tokens at each stage, PVTv2 significantly lowers computational complexity for high-resolution inputs and explicitly models multi-scale features. Benchmarking against PVTv2 helps assess whether our physics-informed time-scaling offers advantages over fixed structural pyramids.

- **ResNeSt** (Zhang et al., 2022) represents a strong multi-branch CNN baseline. It introduces a Split-Attention block that enables feature interaction across different feature map groups, effectively capturing multi-scale visual patterns and dependencies. As a highly competitive CNN variant, ResNeSt serves as a robust baseline to evaluate the efficacy of our proposed method against advanced convolutional attention mechanisms.

By comparing DF-Mamba against these inherent multi-scale architectures, we aim to demonstrate that our performance gains stem from the specific *scale-invariant* inductive bias introduced by the Neural Renormalization Group Flow, rather than simply from the presence of a hierarchical structure.

As shown in Table 8, DF-Mamba consistently outperforms all hierarchical baselines in the zero-shot scale extrapolation setting ($L_{train} = 256 \to L_{test} = 1024$). Specifically, despite Swin Transformer's explicit hierarchical design, it achieves an accuracy of only 64.44% at the target resolution, significantly lower than DF-Mamba's 77.78%. Notably, Swin Transformer requires 27.5M parameters—approximately $30\times$ more than DF-Mamba (0.89M)—yet fails to generalize effectively to the unseen global scale. This suggests that Swin's fixed window-based hierarchy may overfit to the training resolution, whereas DF-Mamba's recursive time-scaling learns a more universal, scale-invariant operator. Similarly, DF-Mamba surpasses PVTv2 (71.11%) and ResNeSt (68.89%) by a clear margin of 6.6% and 8.9%, respectively. While PVTv2 utilizes a pyramid structure to reduce complexity and ResNeSt employs split-attention for multi-scale feature interaction, both still rely on learned scale-specific parameters at each stage. In contrast, DF-Mamba's parameter sharing across recursion depths enforces a consistent physical law, enabling robust extrapolation to larger spatial extents without parameter redundancy. In terms of efficiency, DF-Mamba remains the most lightweight model with the lowest FLOPs (0.72G at $L = 256$), validating that our RG-inspired design achieves superior scale generalization efficiency compared to standard structural hierarchies.

A.10.6. ADDITIONAL CROSS-DOMAIN VALIDATION ON REMOTE SENSING (NWPU-RESISC45)

We provide an additional evaluation on NWPU-RESISC45 (Cheng et al., 2017) to verify DF-Mamba's robustness under scale shifts in a non-medical vision domain. Remote sensing scenes exhibit pronounced multi-scale structure: discriminative cues arise from both fine-grained local textures and global spatial layouts (such as airport, forest, island). This property is analogous to pathology images, where class identity depends jointly on local morphology and tissue-level organization, making remote sensing an appropriate domain to assess DF-Mamba's RG-inspired hierarchical modeling.

We follow the same local-to-global scale extrapolation protocol as in our pathology experiments. We adopt a stratified split with 10% training, 45% validation, and 45% test.

Beyond ViT, ResNet, and flat Mamba (ViM), we additionally compare against three widely-recognized strong multi-scale architectures: **Swin Transformer** and **PVTv2** represent hierarchical/pyramidal Transformers with explicit token downsampling and multi-stage feature hierarchies, while **ResNeSt** is a strong multi-branch CNN that captures multi-scale

*Table 10.* **Hyperparameter search space for sensitivity analysis.**

| HYPERPARAMETER | TESTED VALUES (SEARCH SPACE) |
|---|---|
| EMBEDDING DIM ($D$) | $\{16, 32, 64, 128, 256\}$ |
| SSM STATE DIM ($N$) | $\{8, 16, 32\}$ |
| EXPANSION FACTOR ($E$) | $\{1, 2, 4\}$ |
| CONV KERNEL SIZE | $\{2, 4, 8\}$ |
| FRACTAL DEPTH ($K$) | $\{2, 3, 4\}$ |
| MIN LENGTH STOP | $\{4, 8, 16\}$ |
| $\lambda_{1,2,3}$ | $\{0, 0.1, 0.5, 1\}$ |

patterns via split-attention and has been competitive on remote sensing recognition. These baselines are conceptually closest to our setting because they also aim to integrate local details with global context through explicit hierarchy, thus providing a stricter and fairer test of whether DF-Mamba's recursive, parameter-shared coarse-graining offers additional benefits.

Table 9 reports the extrapolation accuracy. DF-Mamba achieves the best performance in both regimes, obtaining 43.73% for $L_{train}$=64 and 70.30% for $L_{train}$=256, substantially outperforming ViT, ResNet, and flat Mamba (ViM) baselines as well as the strong hierarchical baselines Swin and PVTv2. The margin is particularly large for $L_{train}$=64, where the model must infer global scene semantics from very limited local context, highlighting the benefit of DF-Mamba's recursive coarse-graining and gated multi-scale fusion.

We further analyze two key components. Removing the auxiliary consistency losses reduces accuracy to 37.87% (in $L_{train}$=64) / 67.52% (in $L_{train}$=256), indicating these constraints improve robustness under scale shifts. Disabling the exponential time/scale calibration yields 33.48% (in $L_{train}$=64) / 65.37% (in $L_{train}$=256), confirming that scale calibration is crucial for stable cross-scale modeling.

Overall, the NWPU-RESISC45 results provide additional cross-domain evidence that DF-Mamba learns more scale-stable representations under explicit resolution shifts. These findings support our central claim that RG-inspired recursive coarse-graining, coupled with scale calibration and consistency regularization, enables reliable generalization across resolutions beyond the pathology domain.

### A.11. Hyperparameter Tuning and Analysis

This section summarizes our hyperparameter tuning strategy and the final configurations used in all main experiments. We conducted extensive sensitivity analyses over the search space described in Table 10, where each key hyperparameter of DF-Mamba was systematically varied while fixing others to their optimal values derived from preliminary runs.

The final hyperparameters for the three complementary domains were selected based on validation performance and are reported in Table 11. A deliberate exception was made for the **DF-Mamba-ST** model: to ensure a principled comparison with baselines and align with the biological data structure, its embedding dimension was fixed at 3,000. This configuration matches the input feature dimension of the top-3,000 highly variable genes (HVGs) used in standard spatial transcriptomics benchmarks, preventing information loss from aggressive projection. Conversely, for **DF-Mamba-TS** and **DF-Mamba (Pathology)**, the dimensions were tuned to balance parameter efficiency and representational capacity (64 and 256, respectively). Notably, the *Fractal Depth* adapts to the complexity of the task, with the computational pathology task requiring a deeper recursive hierarchy ($K = 4$) to capture cross-scale tissue organization.

**Sensitivity to fractal depth $K$.** Table 12 reports a sensitivity study over the maximum recursion depth $K \in \{2, 3, 4\}$. On long-term forecasting, the best performance is achieved at $K = 3$ (MSE 0.2318, MAE 0.3234). A shallower hierarchy ($K = 2$) under-exploits multi-scale context and yields higher errors, indicating that at least three scales are beneficial for capturing macroscopic trends. Increasing the depth further to $K = 4$ slightly degrades forecasting accuracy, which is consistent with over-coarsening/over-damping at the coarsest scale when the effective time step grows as $\Delta_k \propto 2^k$.

In contrast, the pathology task benefits from a deeper hierarchy: accuracy improves from 71.11% (for $K = 2, 3$) to 77.78% at $K = 4$. This suggests that histopathology classification relies more heavily on cross-scale tissue organization, where an additional coarse-graining stage can aggregate higher-level morphological patterns beyond local texture cues. Overall, DF-Mamba is robust to moderate changes in $K$, and we select $K = 3$ for forecasting and $K = 4$ for pathology in all main experiments.

*Table 11.* **Final hyperparameter configurations for each domain.** Note that the embedding dimension for DF-Mamba-ST is fixed at 3,000 to match the gene expression input dimension.

| HYPERPARAMETER | DF-MAMBA-TS (TIME SERIES) | DF-MAMBA-ST (SPATIAL) | DF-MAMBA(PATHOLOGY) (PATHOLOGY CLASSIFICATION) |
|---|---|---|---|
| EMBEDDING DIM | 64 | 3000* | 256 |
| SSM STATE DIM | 16 | 16 | 16 |
| EXPANSION FACTOR | 2 | 2 | 2 |
| CONV KERNEL | 4 | 4 | 4 |
| FRACTAL DEPTH | 3 | 3 | 4 |
| MIN LENGTH STOP | 8 | 8 | 8 |
| $\lambda_1$ | 1 | 1 | 1 |
| $\lambda_2$ | 1 | 1 | 1 |
| $\lambda_3$ | 0.1 | 0.1 | 0.1 |

*SET TO MATCH BASELINES FOR FAIR COMPARISON, NOT TUNED.

*Table 12.* **Sensitivity to fractal depth** $K$**.** Lower is better for forecasting (MSE/MAE), while higher is better for pathology (ACC).

| $K$ | MSE$\downarrow$ | MAE$\downarrow$ | ACC (PATHOLOGY)$\uparrow$ |
|---|---|---|---|
| 2 | 0.2444 | 0.3341 | 71.11% |
| 3 | **0.2318** | **0.3234** | 71.11% |
| 4 | 0.2364 | 0.3326 | **77.78%** |

## A.12. Additional Zero-Shot Cross-Scale Comparisons with Strong Non-DF Baselines

To further evaluate whether the proposed DF-Mamba improves robustness to unseen scale shifts, we compare it against the strongest non-DF baseline from our main experiments in each domain. Specifically, we use Affirm for time-series forecasting, DiffusionST for spatial transcriptomics, and PVTv2 for histopathology. All models are trained only at the native scale and directly evaluated on coarsened inputs at $\times 2$ and $\times 4$ scales, without architectural modification, scale-specific fine-tuning, or exposure to the target scale during training.

*Table 13.* Zero-shot cross-scale evaluation on spatial transcriptomics sample 151673. Higher ARI and NMI indicate better performance.

| Scale | DF-Mamba-ST | | DiffusionST | |
|---|---|---|---|---|
| | ARI$\uparrow$ | NMI$\uparrow$ | ARI$\uparrow$ | NMI$\uparrow$ |
| Standard | 0.53 | 0.69 | 0.54 | 0.73 |
| $\times 2$ | 0.46 | 0.61 | 0.28 | 0.46 |
| $\times 4$ | 0.42 | 0.58 | 0.30 | 0.48 |

As shown in Tables 13–15, DF-Mamba achieves competitive performance at the native scale and degrades substantially more slowly under unseen temporal or spatial scale shifts. This trend is consistent across all three domains. In spatial transcriptomics, DiffusionST is slightly stronger at the native scale, but DF-Mamba-ST remains markedly more robust after coarsening, improving ARI from 0.28 to 0.46 at $\times 2$ and from 0.30 to 0.42 at $\times 4$. Similarly, on ETTm2, the MSE increase from the native scale to $\times 4$ is much smaller for DF-Mamba-TS than for Affirm, increasing from 0.158 to 0.192 compared with 0.167 to 0.305. In histopathology, DF-Mamba also retains higher accuracy than PVTv2 under both $\times 2$ and $\times 4$ scale shifts.

These results support the central hypothesis that the shared-operator and coarse-graining-based design of DF-Mamba encourages scale-consistent dynamics, rather than fitting scale-specific patterns at the training resolution only.

## A.13. Ablation on the Coarse-Graining Operator

To assess the contribution of the proposed learned coarse-graining operator, we conduct an ablation study on ETTm2 by replacing the learned pairwise downsampler with three alternative downsampling strategies: max pooling, stride-2 convolution, and spectral low-pass filtering. All other model components and training settings are kept unchanged.

*Table 14.* Zero-shot cross-scale evaluation on ETTm2 time-series forecasting. Lower MSE and MAE indicate better performance.

| Scale | DF-Mamba-TS | | Affirm | |
|---|---|---|---|---|
| | MSE↓ | MAE↓ | MSE↓ | MAE↓ |
| Standard | 0.158 | 0.261 | 0.167 | 0.260 |
| ×2 | 0.173 | 0.275 | 0.251 | 0.314 |
| ×4 | 0.192 | 0.298 | 0.305 | 0.349 |

*Table 15.* Zero-shot cross-scale evaluation on histopathology classification. Higher accuracy indicates better performance.

| Scale | DF-Mamba | PVTv2 |
|---|---|---|
| Standard | 80.60 | 80.60 |
| ×2 | 77.78 | 71.11 |
| ×4 | 71.11 | 62.22 |

As shown in Table 16, convolutional and spectral downsampling provide competitive alternatives, but the proposed learned coarse-graining operator consistently achieves the best performance. Compared with stride-2 convolution, it reduces MSE from 0.167 to 0.158 and MAE from 0.270 to 0.261. The advantage is larger compared with max pooling, which suggests that naive local selection may discard useful temporal information.

These results indicate that the improvements of DF-Mamba do not arise solely from recursive multi-scale processing, but also from the information-preserving design of the learned coarse-graining operator. This supports our motivation that effective scale construction is important for maintaining predictive dynamics across resolutions.

### A.14. Analysis of Clipping at Deeper Scales

In DF-Mamba, the scale-dependent discretization step is defined as $\Delta_k = \Delta_0 \cdot 2^k$, which follows from discretization consistency under a shared continuous-time state-space operator. In practice, we apply clipping to the scaled discretization step as an implementation-level safeguard to avoid numerical instability at very coarse scales. Importantly, this clipping operation is not part of the theoretical construction of the recursive SSM, but only a stability mechanism used during training and inference.

To quantify its effect, we conduct an explicit analysis on ETTh2. We first measure how frequently clipping is activated at each recursive depth when the clipping threshold is set to $8.0$. As shown in Table 17, clipping is inactive at shallow levels, with a clipping rate of $0\%$ at $k = 0$ and $k = 1$. It is only mildly activated at $k = 2$ with a rate of $8.9\%$, and becomes frequent mainly at the deepest level $k = 3$, where the clipping rate reaches $60.2\%$. This suggests that clipping mainly acts as a safeguard for coarse-scale dynamics, rather than dominating the behavior of the hierarchy.

We further evaluate the sensitivity of DF-Mamba to different clipping thresholds. As shown in Table 18, the test MSE remains stable across a broad reasonable range of thresholds. Specifically, the test MSE is $0.2319$ with threshold $4.0$, $0.2322$ with threshold $8.0$, and $0.2323$ with threshold $16.0$. Disabling clipping slightly worsens the result to $0.2326$.

Overall, these results show that DF-Mamba is not strongly sensitive to the precise clipping threshold within a reasonable range. Clipping primarily improves numerical stability at the deepest recursive levels, while the model's performance is mainly driven by the shared continuous-time operator and the recursive coarse-graining hierarchy.

### A.15. Limitations and Future Work

While DF-Mamba establishes a rigorous framework for physics-inspired sequence modeling, we identify specific limitations that open avenues for future research.

As observed in our computational pathology analysis, DF-Mamba exhibits high stability in capturing topological invariants but struggles with semantic boundary cases. Specifically, the model consistently misclassified the hard InSitu sample as Normal across both coarse and fine scales. This suggests that while the RG flow effectively filters noise to extract global structure, the current deterministic coarse-graining might inadvertently smooth over subtle, hyper-local textural anomalies

*Table 16.* Ablation study of different coarse-graining operators on ETTm2. Lower MSE and MAE indicate better performance.

| Coarse-graining operator | MSE↓ | MAE↓ |
|---|---|---|
| Learned pairwise downsampler (ours) | 0.158 | 0.261 |
| Stride-2 convolution | 0.167 | 0.270 |
| Spectral low-pass filtering | 0.171 | 0.269 |
| Max pooling | 0.176 | 0.274 |

*Table 17.* Clipping activation frequency across recursive depths on ETTh2 using a clipping threshold of 8.0.

| Depth $k$ | 0 | 1 | 2 | 3 |
|---|---|---|---|---|
| Clipping rate (%) | 0.0 | 0.0 | 8.9 | 60.2 |

that define transitional disease states.

To resolve ambiguities in boundary cases (such as InSitu vs. Normal), we plan to incorporate cross-scale contrastive learning. By explicitly penalizing the feature similarity between hard-negative pairs at the fine-grained level while maintaining consistency at the coarse level, we can encourage the model to preserve critical high-frequency residuals that define transitional states. Beyond specific tasks, DF-Mamba serves as a proof-of-concept for embedding physical inductive biases into deep learning. We aim to scale this architecture to train multi-modal scientific foundation models, capable of seamlessly processing diverse hierarchical data, from genomic sequences to climate patterns, within a unified, scale-invariant framework.

*Table 18.* Sensitivity analysis of different clipping thresholds on ETTh2. Lower MSE indicates better performance.

| Clipping threshold | 4.0 | 8.0 | 16.0 | No clipping |
|---|---|---|---|---|
| Test MSE↓ | 0.2319 | 0.2322 | 0.2323 | 0.2326 |

---

**Algorithm 1** Dynamic Fractal Mamba (DF-Mamba)

---

**Require:** Input sequence $\mathbf{X} \in \mathbb{R}^{B \times L \times D}$, recursion depth $k$
**Require:** Max depth $K$, minimum length $L_{\min}$
**Require:** Shared Mamba blocks $\mathcal{S}_\theta$ (with internal $\Delta(\cdot)$ prediction)
**Require:** Downsampler $\mathcal{D} = (\text{Enc}, \text{Dec})$, Upsampler $\mathcal{U}$, Gate network $\mathcal{G}$
**Ensure:** Output features $\mathbf{H}_{out}$, accumulated auxiliary loss $\mathcal{L}_{aux}$
 1: **function** RECURSIVEFORWARD($\mathbf{X}, k$)
 2:     **// 1. Depth-dependent time scaling**
 3:     $\alpha \leftarrow 2^k$          ▷ RG intuition: exponential rescaling of effective time step
 4:     **// 2. Local sequence modeling (Mamba)**
 5:     $\mathbf{H}_{local} \leftarrow \mathbf{X} + \mathcal{S}_\theta(\text{Norm}(\mathbf{X}); \Delta(\mathbf{X}) \cdot \alpha)$        ▷ Local dynamics (microscopic evolution)
 6:     **// 3. Base case**
 7:     **if** $k \geq K$ **or** $L < L_{\min}$ **then**
 8:         **return** $\mathbf{H}_{local}, 0, \text{Null}$
 9:     **end if**
10:     **// 4. Information-preserving downsampling**
11:     $\mathbf{P} \leftarrow \text{PAIRCONCAT}(\mathbf{H}_{local})$
12:     $\mathbf{X}_{next} \leftarrow \text{Enc}(\mathbf{P})$        ▷ RG intuition: block-spin / coarse-graining
13:     $\hat{\mathbf{P}} \leftarrow \text{Dec}(\mathbf{X}_{next})$
14:     $\mathcal{L}_{recon} \leftarrow \|\hat{\mathbf{P}} - \text{sg}(\mathbf{P})\|_2^2$        ▷ Information conservation across scales
15:     **// 5. Recursive multi-scale processing**
16:     $\mathbf{H}_{deep}, \mathcal{L}_{inner}, m_{next} \leftarrow \text{RECURSIVEFORWARD}(\mathbf{X}_{next}, k+1)$
17:     **// 6. Resolution recovery**
18:     $\mathbf{H}_{global} \leftarrow \mathcal{U}(\mathbf{H}_{deep}; \text{size} = L)$        ▷ RG intuition: scale inversion
19:     **// 7. Cross-scale consistency**
20:     $\mathcal{L}_{consist} \leftarrow \|\mathbf{H}_{global} - \text{sg}(\mathbf{H}_{local})\|_2^2$        ▷ Approximate fixed-point consistency
21:     **// 8. Adaptive fusion**
22:     $\mathbf{g} \leftarrow \sigma(\mathcal{G}([\mathbf{H}_{local}, \mathbf{H}_{global}]))$
23:     $\mathbf{H}_{out} \leftarrow (1 - \mathbf{g}) \odot \mathbf{H}_{local} + \mathbf{g} \odot \mathbf{H}_{global}$        ▷ Learned balance of context
24:     **// 9. Scale-wise gate smoothness**
25:     $m_k \leftarrow \text{mean}_{b,l,d}(\mathbf{g})$        ▷ Mean over batch/length/(optional channel) dimensions
26:     $\mathcal{L}_{flow} \leftarrow 0$
27:     **if** $m_{next} \neq \text{Null}$ **then**
28:         $\mathcal{L}_{flow} \leftarrow \|m_k - \text{sg}(m_{next})\|_2^2$
29:     **end if**
30:     **// 10. Accumulate auxiliary loss**
31:     $\mathcal{L}_{aux} \leftarrow \mathcal{L}_{inner} + \lambda_1 \mathcal{L}_{recon} + \lambda_2 \mathcal{L}_{consist} + \lambda_3 \mathcal{L}_{flow}$
32:     **return** $\mathbf{H}_{out}, \mathcal{L}_{aux}, m_k$
33: **end function**

---

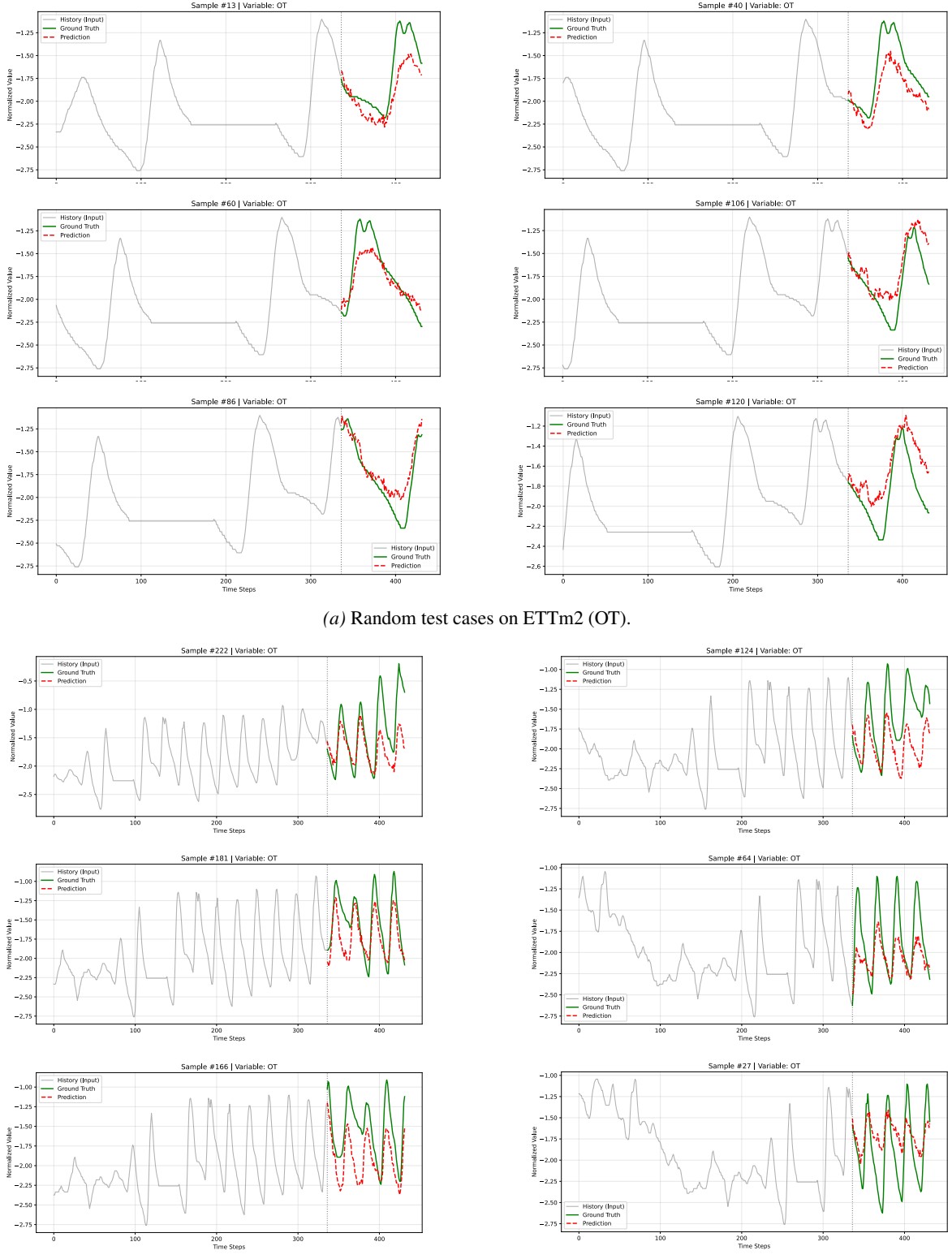

*(a)* Random test cases on ETTm2 (OT).

*(b)* Random test cases on ETTh2 (OT).

*Figure 4.* **Random qualitative forecasts on ETT benchmarks (OT).** Each panel shows the normalized OT series with history input (gray), ground-truth future (green), and model prediction (red dashed). The vertical dashed line indicates the boundary between look-back ($L = 336$) and prediction horizon ($H = 96$). (a) and (b) correspond to two ETT benchmarks (ETTm2 / ETTh2).

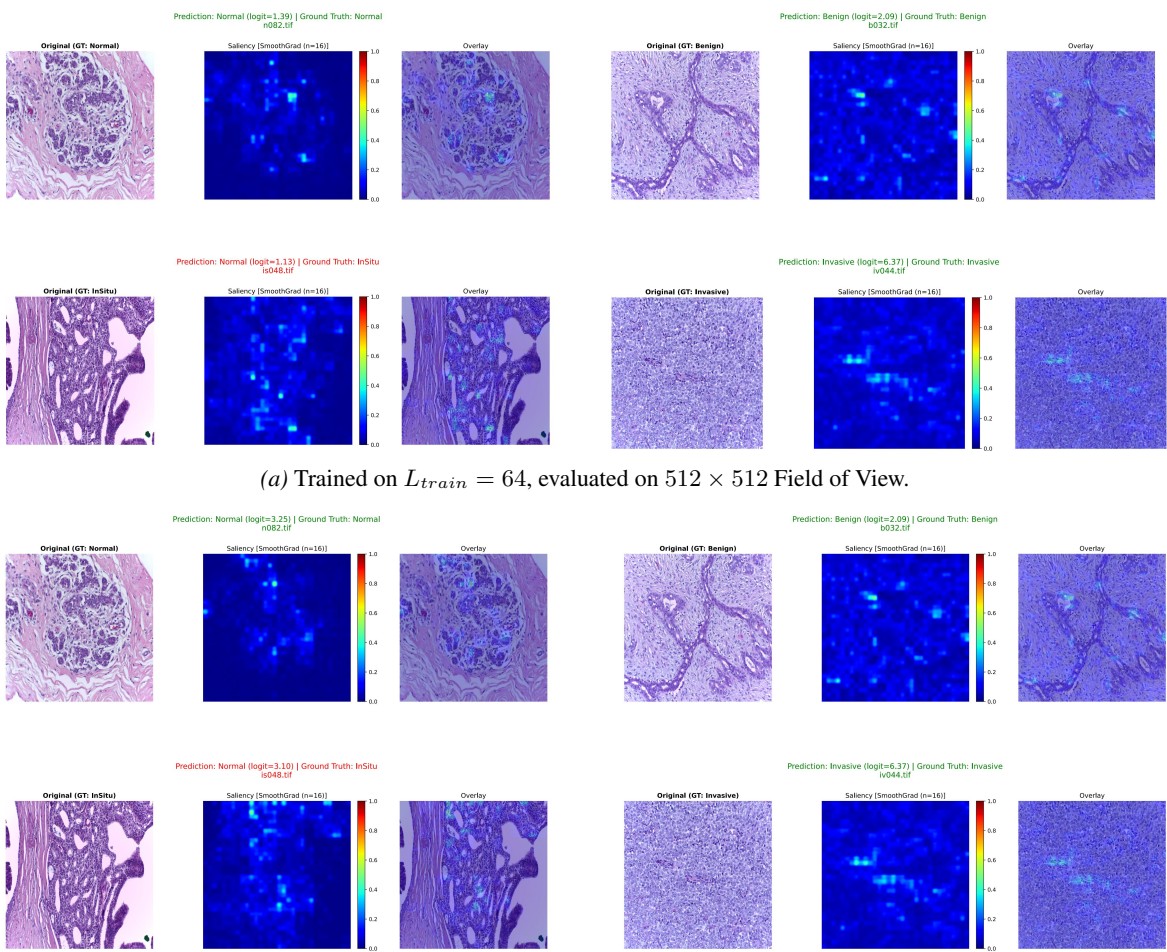

*(a)* Trained on $L_{train} = 64$, evaluated on $512 \times 512$ Field of View.

*(b)* Trained on $L_{train} = 256$, evaluated on $512 \times 512$ Field of View.

*Figure 5.* **Pathology interpretability under scale shift.** We visualize pixel-level evidence supporting DF-Mamba's predictions using input-gradient saliency. For each example, the **left** panel shows the original pathology image, the **middle** panel shows the gradient saliency map for the predicted class, normalized to $[0, 1]$, and the **right** panel overlays the saliency heatmap on the original image. The visualization is reported under cross-scale evaluation: models trained on smaller crops are tested on a larger field-of-view ($512 \times 512$) to assess whether the model focuses on scale-consistent morphological cues.

