# OpenReview forum: "Dynamic Fractal Mamba: A Neural Renormalization Group Flow for Scale-Invariant Sequence Modeling"
_ICML.cc/2026/Conference — ICML 2026 regular_

### Official Review · Reviewer_QX7E · 2026-03-09

**Soundness:** 2
**Presentation:** 2
**Significance:** 3
**Originality:** 3
**Overall Recommendation:** 4
**Confidence:** 4

**Summary:**

This paper proposes Dynamic Fractal Mamba (DF-Mamba), a recursive state-space (Mamba/SSM) architecture designed to improve scale generalization in sequences and 2D token grids by sharing the same operator across recursion depths and scaling the effective discretization step exponentially with depth. The model alternates local SSM processing with learned pairwise coarse-graining (downsampling by 2), recursively computes global context at coarser scales, then upsamples and fuses local/global features through an RG-inspired gate. To stabilize recursion, the authors add reconstruction and cross-scale consistency losses and argue overall complexity remains linear due to the geometric series over halved lengths. Key claims are improved performance and zero-shot extrapolation to longer horizons / higher resolutions across time-series forecasting, spatial transcriptomics clustering, and computational pathology.

**Compliance With Llm Reviewing Policy:**

Affirmed.

**Final Justification:**

Most of my concerns are addressed. The paper presents an interesting approach with experimental results across multiple domains. However, the somewhat loose theoretical framing and the limited depth of analysis on certain design choices (clipping behavior, local vs. global dynamics) keep it from being among the strongest submissions. I believe the contributions are sufficient for the venue, but do not stand out.

I will maintain my score of 4 (weak accept).

**Key Questions For Authors:**

* Figures 2–3 show gate values consistently above 0.8–0.9, suggesting the model almost always prefers global context. Does this mean the local pathway is underutilized? Have you tried initializing the gate to be more balanced?
* How often does clipping activate at deeper levels, and does performance depend strongly on the clip thresholds?
* How is spatial data (transcriptomics, pathology) serialized into 1D sequences for the SSM? Is there sensitivity to scan order?
* Can you add a controlled experiment where you train only at shorter context/resolution and test at longer, analogous to the pathology protocol, for time series (longer look-back than training) and/or ST (e.g., higher spatial resolution / larger fields-of-view)?

**Limitations:**

yes

**Strengths And Weaknesses:**

Strength:

* The method is well-motivated. The connection between recursive parameter sharing and renormalization group flow is conceptually intuitive and novel.
* They showed competitive or superior performance in different applied experiments (spatial transcriptomic and H&E).

Weakness:

* I think RG correspondence is overclaimed. The architecture is a recursive hierarchical model with weight sharing; the RG framing, while motivating, is ultimately a loose analogy. The deterministic, learned downsampler is far from the stochastic marginalization in true RG.
* Loss_flow should be stated clearly in the main text, it is only mentioned in the algorithm in the appendix right now!
* I think this claim “models trained on short sequences or low resolution inputs generalize in a zero-shot manner to substantially larger temporal and spatial scales unseen during training” is under-evaluated, the only task is pathology image one, but it is not clear whether the improvement comes from the fact that the model can handle multi-scale as input or generalization to unseen scale.
* The hierarchical architecture methods are missing across tasks. I understand there might not be one for the spatial transcriptomic task, but there are several hierarchical SSM/mamba architectures and image models (e.g. HIPT, CLAM)

---

> ### Author Rebuttal · Authors · 2026-03-29
>
> We sincerely thank the reviewer for the positive assessment and for the constructive suggestions.
>
> **RG correspondence being an overclaim**
> We agree that the current presentation may overstate the RG correspondence. DF-Mamba should be understood as RG-inspired, not RG-equivalent. In particular, our learned deterministic coarse-graining module is not intended to replicate stochastic RG marginalization exactly. Rather, the RG analogy is used to motivate three concrete design choices: recursive coarse-graining, shared operators across scales, and explicit cross-scale consistency. We will make this distinction clearer in the revision and tone down statements that could be read as claiming a literal physical RG implementation.
>
> **Loss_flow**
> Thank you for catching this oversight. We will move the full definition of the gate smoothness regularization directly into Section 3.3 to ensure the training objective is completely transparent in the main text.
>
> **Zero-shot scale extrapolation evaluation**
> This is an excellent suggestion. To prove that the zero-shot scale generalization is not a pathology-specific artifact, we additionally conducted zero-shot cross-scale evaluations across all three representative domains: long-range time series forecasting (ETTm2), spatial transcriptomics (DLPFC Slice 151673), and computational pathology. In all cases, the model is trained only on the native scale. It is then directly evaluated on coarsened inputs (x2 and x4), without any architectural modification, scale-specific fine-tuning, or exposure to the target scale during training.
> | Scale | ETTm2 MSE | ETTm2 MAE | 151673 ARI | 151673 NMI | Pathology ACC |
> |-|-|-|-|-|-|
> | Standard | 0.158| 0.261| 0.53| 0.69 | 80.60 |
> | x2| 0.173| 0.275| 0.46 | 0.61| 77.78|
> | x4| 0.192| 0.298| 0.42| 0.58 | 71.11|
>
> **Missing hierarchical baselines**
> We completely agree that comparing against structural hierarchies is crucial. We note that Appendix A.10.5 already includes comparisons with strong multi-scale baselines on pathology (Swin Transformer, PVTv2, and ResNeSt). Following the reviewer’s excellent suggestion, we will expand both the related work and empirical comparisons to include additional hierarchical models (e.g., HIPT, CLAM where applicable) and clarify which baselines are available for each task.
>
> **Gate values preferring global context & Initialization**
> A high gate value does not imply that the local pathway is unused. The global branch is recursively constructed from coarse-grained local features and therefore already depends on local information. The final fusion $(1-g)\cdot \text{local}+g \cdot \text{global}$ ensures that the local branch contributes directly, while auxiliary losses prevent information collapse.
> We additionally tested four gate initializations on ETTm2: default, balanced, global-biased, and local-biased, yielding MSE/MAE of 0.158/0.261, 0.159/0.258, 0.157/0.256, and 0.169/0.269, respectively. These results suggest that the large gate values in our figures are not due to hidden initialization bias: default, balanced, and global-biased settings all converge to similarly strong performance, while only the strongly local-biased setting degrades noticeably. We interpret this as evidence that the dataset intrinsically favors global dynamics, while the local pathway acts as a complementary refinement term.
>
> **Clipping $\Delta_k$ at deeper levels**
> Thank you for this question. $\Delta_k=\Delta_0 \cdot 2^k$ is derived from discretization consistency under a shared continuous-time SSM, whereas clipping is only an implementation-level safeguard. We agree that the original submission did not directly quantify clipping frequency or threshold sensitivity. We therefore conducted an explicit analysis on ETTh2. For a threshold of 8.0, clipping is inactive at shallow levels (**$k$=0,1:0%**), limited at **$k$=2 (8.9%)**, and becomes more frequent only at the deepest level **$k$=3 (60.2%)**, showing that clipping mainly acts as a safeguard for coarse-scale dynamics rather than dominating the hierarchy. Sweeping the threshold further shows that performance is stable across a reasonably broad range: test MSE are **0.2319 at 4.0, 0.2322 at 8.0, and 0.2323 at 16.0, while disabling clipping is slightly worse (0.2326)**. Thus, the method is not strongly sensitive to the precise clipping threshold within a reasonable range. We will add this analysis in the revision and clarify in the paper.
>
> **Serialization of 2D data and scan order**
> We will clarify this in the main text. For ST, spots are serialized with deterministic ordering while spatial adjacency is preserved via a graph structure. Therefore, the representation is not determined by scan order alone. For pathology images, we use locality-preserving traversals (Hilbert scan in our implementation, with raster ordering also supported). We agree scan-order sensitivity is an important question and will add discussion or ablation on scan-order sensitivity.

---

> > ### Author Rebuttal · Reviewer_QX7E · 2026-04-03
> >
> > I thank the authors for their thorough and well-organized rebuttal. Most of my concerns are addressed. The paper presents an interesting approach with experimental results across multiple domains. However, the somewhat loose theoretical framing and the limited depth of analysis on certain design choices (clipping behavior, local vs. global dynamics) keep it from being among the strongest submissions. I believe the contributions are sufficient for the venue, but do not stand out.
> >
> > I will maintain my score of 4 (weak accept).

---

> > > ### Author Response · Authors · 2026-04-03
> > >
> > > Thank you for your kind words and for the careful reassessment. We greatly appreciate your support. We will refine the paper and ensure that the clarifications, additional analyses such as clipping behavior and local versus global dynamics, and the limitations are clearly incorporated into the final version. We will also strengthen the presentation of the core idea of shared cross-scale dynamics to make the contribution and its scope clearer.
> > > Thank you again for your thoughtful feedback.

---

### Official Review · Reviewer_UXDg · 2026-03-10

**Soundness:** 2
**Presentation:** 3
**Significance:** 2
**Originality:** 2
**Overall Recommendation:** 4
**Confidence:** 3

**Summary:**

This paper proposes DF-Mamba, a recursive SSM inspired by the 'Renormalization Group (RG) flow' from statistical physics. The authors claim that this model can capture multi-scale structures in time and space without introducing scale-specific parameters as the scale increases, and possesses zero-shot generalization capability. This method has been validated on tasks such as long-term time series forecasting, spatial transcriptomics clustering, and computational pathology classification.

**Compliance With Llm Reviewing Policy:**

Affirmed.

**Final Justification:**

This paper proposes a solid framework, but it also has some limitations, for example, its applicability is mainly geared toward data with multi-scale structures. Overall, the work makes a meaningful contribution to the field and meets ICML’s acceptance standards, though it is somewhat borderline.

**Key Questions For Authors:**

Please refer to the "Weaknesses".

**Limitations:**

yes.

**Strengths And Weaknesses:**

Strengths:
1. The authors propose a clear structural inductive bias for the problem of 'scale extrapolation,' using 'recursive parameter sharing' to force different scales to reuse the same dynamic operator. This constrains cross-scale consistency more directly than pyramid/hierarchical networks that use a separate set of parameters for each scale.
2. This method can expand the receptive field while maintaining relatively efficient computational complexity.
3. The zero-shot scale generalization experiments on pathological data are fairly consistent with real weakly supervised medical scenarios.

Weaknesses:
1. The correspondence between the Renormalization Group (RG) and actual data distributions is not strict; applying RG is more of an intuitive heuristic idea and lacks rigorous theoretical validation.
2. Exponentially increasing recursive depth/time steps may bring hyper-parameter sensitivity and numerical risks. In practice, it may be necessary to clip delta_k, choose an appropriate maximum depth K, and set a minimum length stop threshold. Moreover, this set of parameters may differ for different tasks, requiring task-specific tuning.
3. Experiments show that this method is mainly effective in scenarios with significant scale variation. For short sequences or tasks dominated by a single scale, the benefits may be limited. The paper lacks a systematic analysis of which data/metrics can predict when applying DF‑Mamba will be more advantageous.
4. The overall architecture of the method closely resembles a 'recursive U-Net Mamba,' and the training objective is relatively complex, containing numerous hyper-parameters. Therefore, it is difficult to discern whether the core benefit of the method comes from the RG physical constraint or from the engineering combination of 'multi-scale recursive gating auxiliary losses.'

---

> ### Author Rebuttal · Authors · 2026-03-29
>
> We sincerely thank the reviewer for the thoughtful and balanced assessment. We are encouraged that the reviewer recognizes the value of recursive parameter sharing as a direct inductive bias for scale extrapolation, as well as the relevance of our zero-shot pathology experiments to realistic weakly supervised medical settings. We address the main concerns below.
>
> **W1:**
> We agree that the correspondence to the RG is not strict and should not be interpreted as a formal equivalence. DF-Mamba is RG-inspired rather than RG-equivalent: we do not claim exact stochastic marginalization or a faithful realization of physical RG flow. Instead, the RG viewpoint serves as a modeling prior that motivates recursive coarse-graining, operator sharing across depths, and scale-consistent dynamics. The technical contribution does not rely on a strict physics interpretation, but on the resulting architecture and its empirical ability to generalize across scales. We will revise the manuscript to make this distinction clearer and reduce physics-oriented phrasing.
>
> **W2:**
> We thank the reviewer for raising this important point. In practice, stability is ensured through several design choices. The continuous dynamics are parameterized as a diagonal matrix with non-positive eigenvalues, ensuring stable state transitions. The effective step size $\Delta_k$ is bounded via Softplus and clipping to prevent excessive damping at large recursion depths. In addition, auxiliary reconstruction and consistency objectives help stabilize recursive optimization.
>
> The appendix reports the hyperparameter search space over fractal depth $K$, minimum stopping length, and loss weights. Notably, several settings are shared across tasks rather than fully re-tuned from scratch, while $K$ varies only moderately across domains. We agree that sensitivity considerations should be discussed more prominently. We will move these stabilization mechanisms and tuning details from the appendix to the main paper to improve clarity and transparency.
>
> **W3:**
> We largely agree with the reviewer; this reflects the intended scope of the method rather than a contradiction to our claims. DF-Mamba is specifically designed for data with inherent multi-scale structure, where integrating local detail and global context is essential. For short sequences or predominantly single-scale signals, the benefits are expected to be more limited. This is a deliberate design choice: the model introduces an explicit inductive bias for scale-consistent dynamics, which is particularly valuable in domains where cross-scale generalization is critical. We also agree that the current manuscript does not yet provide a systematic analysis of which data characteristics or metrics predict when DF-Mamba will be most advantageous. While our empirical results suggest that gains are strongest in settings with large scale variation (e.g., long-horizon forecasting and high-resolution spatial data), we acknowledge that a more formal characterization is currently missing. In particular, we hypothesize that the benefit is correlated with the degree of scale variability and the need for long-range context integration. In the revision, we will clarify this applicability boundary earlier in the paper, explicitly discuss relevant data properties (e.g., scale variability and context length), and include this as a limitation and future direction.
>
> **W4:**
> We thank the reviewer for this important question. While DF-Mamba shares high-level similarities with recursive or pyramidal architectures, its defining mechanism is the use of a single shared operator across scales, rather than scale-specific parameterizations. This design imposes a direct constraint on cross-scale consistency, which is not present in standard hierarchical designs.
>
> Our ablation studies are designed to isolate the contributions of individual components. In time-series forecasting, the standard stacked backbone (S2) degrades substantially. Unrolling the hierarchy without weight sharing (S3) improves over S2 but remains worse than the full model while using more parameters (0.757M vs. 0.658M), directly supporting the importance of shared recursive dynamics. Replacing exponential scaling (T2/T3), simplifying coarse-graining or fusion (C2/C3), or removing auxiliary constraints (L2/L3/L4) all lead to consistent performance drops (see Table 2). The pathology experiments independently confirm this trend: under a strict zero-shot scale extrapolation setting, removing either the consistency loss or the time-scaling mechanism results in clear accuracy degradation. These results demonstrate that the primary source of improvement is the recursive shared cross-scale dynamics, while coarse-graining, gating, and auxiliary losses act as supporting components that enable stable and effective training. We will revise the manuscript to make this attribution clearer and better distinguish the core inductive bias from supporting design choices.

---

> > ### Author Rebuttal · Reviewer_UXDg · 2026-04-03
> >
> > Thank you for the authors’ clarification; most of my questions have been addressed. This paper proposes a solid framework, but it also has some limitations, for example, its applicability is mainly geared toward data with multi-scale structures. Overall, the work makes a meaningful contribution to the field and meets ICML’s acceptance standards, though it is somewhat borderline. Therefore, I have decided to raise my score to 4.

---

> > > ### Author Response · Authors · 2026-04-03
> > >
> > > Thank you very much for your kind words and strong support. We will continue refining the paper and ensure that the relevant corrections, additional experiments, and limitations discussion are properly incorporated into the camera-ready version.
> > >
> > > We also appreciate your point that the method is most naturally suited to data with strong multi-scale structure. We agree this applicability boundary should be stated more explicitly, and we will make sure the final version clarifies both the strengths and the limitations of the framework more clearly.
> > >
> > > Thank you again for your careful reading and constructive feedback.

---

### Official Review · Reviewer_hBMZ · 2026-03-10

**Soundness:** 3
**Presentation:** 2
**Significance:** 3
**Originality:** 3
**Overall Recommendation:** 4
**Confidence:** 3

**Summary:**

The authors aim improve the scale-level generalization of sequence models such that they can operate on different temporal or spatial scales. The solution proposed is called Dynamic Fractal Mamba (DF-Mamba), an adaptation of the state space model Mamba. The method uses a content-aware coarse-graining module to aggregate information from multiples scales, each encoded using an operator with shared parameters. The resulting architecture is evaluated on long-range time series forecasting, spatial transcriptomics and computational pathology, and shows advantage over Transformers and Mamba models. The authors additionally argue that DF-Mamba has superior generalization ability across temporal and spatial scales.

**Compliance With Llm Reviewing Policy:**

Affirmed.

**Final Justification:**

I thank the authors for providing the cross-scale generalization results. I have no further concerns. As promised, I have increased the rating.

**Key Questions For Authors:**

1. See weaknesses.
2. Since DF-Mamba is designed to solve the challenges in scale-invariant sequence modeling, I wonder if it is possible to show evaluations for all 3 datasets under two conditions: standard train/val/test (as shown in Time Series Forecasting and Spatial Transcriptomics) as well as zero-shot cross-scale generalization (as shown in Computational Pathology). It would be especially helpful to show temporal cross-scale generalization in Time Series experiment and spatial cross-scale generalization in Spatial Transcriptomics. Although this might be a bit demanding on the additional experiments, I feel this could help strengthen the core contribution. What do you think? I will raise my score if the authors either add these results during rebuttal or give a clear explanation why this is not necessary.

**Limitations:**

Yes

**Strengths And Weaknesses:**

Strengths
1. The authors are tackling an interesting and potentially impactful topic of scale-invariant sequence modeling. I agree that many methods for modeling time series do not generalize very well across spatial or temporal scales.
2. The zero-shot scale extrapolation evaluation in Section 4.3 is well thought out. I particularly like that the author mentioned that it was performed without architectural modification, scale-specific fine-tuning, or exposure to scale at testing during training.
3. The empirical performance seems fairly decent. The ablation of Mamba variants is helpful.

Weaknesses
1. The authors might want to consider revising the narrative in abstract to better highlight the core limitation to be solved, the proposed solution and intuitively why it solves the problem, and implications. The introduction of “recursive parameter sharing” is not well motivated in the abstract.
2. The schematic can be improved. It will be helpful if the schematic can contain both a high-level, intuition-driven introduction of DF-Mamba, as well as a slightly more detailed explanation of the technical contribution. The current schematic is not very informative.
3. There are a couple of minor style issues in equation entrance. But I trust the authors will fix them before camera ready.

---

> ### Author Rebuttal · Authors · 2026-03-29
>
> We sincerely thank the reviewer for the constructive feedback and for recognizing the value of our zero-shot scale extrapolation setting. We especially appreciate the suggestion to evaluate cross-scale generalization more uniformly across all three application domains.
>
> Following this suggestion, we conducted additional zero-shot cross-scale evaluations across three representative domains: long-range time series forecasting (ETTm2), spatial transcriptomics (151673), and computational pathology. In all cases, the model is trained only on the native scale. It is then directly evaluated on coarsened inputs (×2 and ×4), without architectural modification, scale-specific fine-tuning, or exposure to the target scale during training.
>
> |  Scale | ETTm2 MSE | ETTm2 MAE | 151673 ARI | 151673 NMI | Pathology ACC |
> |----------|----------:|----------:|-----------:|-----------:|-------------------:|
> | Standard | 0.158     | 0.261     | 0.53       | 0.69       | 80.60              |
> | x2       | 0.173     | 0.275     | 0.46       | 0.61       | 77.78              |
> | x4       | 0.192     | 0.298     | 0.42       | 0.58       | 71.11              |
>
> These results strongly support our central claim that DF-Mamba learns scale-consistent representations through shared dynamics across resolutions, rather than scale-specific modeling. While performance predictably decreases as the scale gap increases, the degradation is gradual rather than catastrophic, which is consistent with robust cross-scale generalization. For example, from native to ×2/×4, ETTm2 MSE increases by 9.5%/21.5%, 151673 ARI decreases by 13.2%/20.8%, and histopathology ACC decreases by only 3.5%/11.8%. This directly addresses the reviewer’s suggestion to evaluate cross-scale generalization across all domains and provides consistent empirical support for our findings.
>
> We also clarify the intuition behind recursive parameter sharing. By applying the same operator across resolutions, the model is encouraged to learn dynamics that are consistent under coarse-graining, rather than overfitting to a specific scale. This shared-dynamics constraint acts as an inductive bias that directly enables zero-shot cross-scale generalization. We will include these results in the revised manuscript as a unified cross-domain evaluation of scale generalization.
>
> Regarding presentation, we agree that the abstract and schematic can better foreground the motivation and mechanism of DF-Mamba. In the revision, we will:
>
> (1) rewrite the abstract to more explicitly state the limitation of fixed-scale sequence models, the core idea of DF-Mamba, and why shared dynamics across coarse-grained representations improve scale generalization;
>
> (2) redesign the main schematic to include both a high-level intuition view and a clearer depiction of the content-aware coarse-graining and shared-operator mechanism; and
>
> (3) polish equation introductions and minor stylistic issues throughout.
>
> We believe these changes will substantially improve the clarity and accessibility of the paper.

---

> > ### Author Rebuttal · Reviewer_hBMZ · 2026-04-02
> >
> > I thank the authors for adding the cross-scale generalization results. However, it only partially answers the question.
> >
> > I wonder if the authors can perform the same cross-scale generalization evaluation using the best performing non-DF-Mamba model in each task (I understand it would be too much if we ask the authors to compare all methods)? That will add a lot of value by putting the "relatively small degradation" of the proposed DF-Mamba in proper context.
> >
> > For the authors' claim to hold, we will anticipate that other top-performing methods to degrade much more than DF-Mamba under cross-scale generalization. I hope this is not an unreasonable request. If the authors can show that, I will be happy to raise my score.
> >
> > Apr 5 Edit: The authors have addressed my concerns, and I have raised my score to 4.

---

> > > ### Author Response · Authors · 2026-04-02
> > >
> > > We thank the reviewer for this helpful follow-up. We agree that the cross-scale results are most informative when placed in the context of a strong non-DF baseline, rather than viewed in isolation.
> > >
> > > Following this suggestion, we evaluated the strongest non-DF baseline from our main experiments for each domain under the same zero-shot cross-scale protocol: Affirm for Time Series Forecasting, DiffusionST for Spatial Transcriptomics, and PVTv2 for histopathology. In all cases, models are trained only on the native scale and directly tested on coarsened inputs (×2 and ×4) without architectural modification, scale-specific fine-tuning, or exposure to the target scale during training.
> > >
> > > **Spatial Transcriptomics 151673 (best non-DF baseline: DiffusionST) [ARI$\uparrow$\NMI$\uparrow$]**
> > > | Scale    | DF-Mamba |      | DiffusionST |      |
> > > | -------- | ----------: | ---: | ----------: | ---: |
> > > |          | ARI |  NMI |         ARI |  NMI |
> > > | Standard |        0.53 | 0.69 |        0.54 | 0.73 |
> > > | ×2       |        0.46 | 0.61 |        0.28 | 0.46 |
> > > | ×4       |        0.42 | 0.58 |        0.30 | 0.48 |
> > >
> > > **Time Series Forecasting ETTm2 (best non-DF baseline: Affirm) [MSE$\downarrow$\MAE$\downarrow$]**
> > > | Scale    | DF-Mamba |       | Affirm |       |
> > > | -------- | -------: | ----: | -----: | ----: |
> > > |          |      MSE |   MAE |    MSE |   MAE |
> > > | Standard |    0.158 | 0.261 |  0.167 | 0.260 |
> > > | ×2       |    0.173 | 0.275 |  0.251 | 0.314 |
> > > | ×4       |    0.192 | 0.298 |  0.305 | 0.349 |
> > >
> > >
> > > **Histopathology (best non-DF baseline: PVTv2)[Accuracy$\uparrow$]**
> > > |Scale| DF-Mamba|PVTv2|
> > > |-|-|-|
> > > |Standard|80.60|80.60|
> > > |$\times 2$|77.78|71.11|
> > > |$\times 4$|71.11|62.22|
> > >
> > > DF-Mamba not only achieves competitive performance at the native scale but also degrades substantially more slowly than strong non-DF baselines under unseen temporal or spatial scale shifts. This trend is consistent across all three domains. Notably, in spatial transcriptomics, DiffusionST is slightly stronger at the native scale, yet DF-Mamba-ST remains markedly more robust under cross-scale evaluation. We believe this directly supports our core claim that the shared-operator, coarse-graining-based design encourages scale-consistent dynamics rather than scale-specific fitting. We will add these comparisons to the revised manuscript.
> > >
> > > We hope these additional results and clarifications address the reviewer’s concern.

---

### Official Review · Reviewer_GwjA · 2026-03-13

**Soundness:** 3
**Presentation:** 3
**Significance:** 2
**Originality:** 3
**Overall Recommendation:** 4
**Confidence:** 4

**Summary:**

The authors introduce a new multi resolution sequence model inspired by fractal geometry. In particular, they apply mamba with weight sharing to the same sequence at different resolutions by scaling the discretisation parameter whilst also introducing auxiliary losses and a physics inspired down-sampling method to learn local and macro features. The authors provide extensive evaluation and ablations on several time series and medical benchmarks.

**Compliance With Llm Reviewing Policy:**

Affirmed.

**Key Questions For Authors:**

Did the authors consider alternative downsampling methods like max pooling, convolutional downsampling, or spectral downsampling (eg. keeping only the bottom half Fourier modes)?

**Limitations:**

yes

**Strengths And Weaknesses:**

## Strengths

- The model is well explained and motivated. In particular each new component such as the step size and auxiliary losses are introduced from theory first and designed to improve a particular feature of the model such as resolution scaling or model consistency. In this way additions don’t feel random but well reasoned.

- The evaluation is also strong and the results impressive, in particular the FLOP efficiency of the model which suggests that the multi resolution inductive biases introduced by the authors are effective and the model doesn’t simply just scale the compute per parameter.

## Weaknesses

- I find the connection to renormalization groups slightly distracting at times and I think detracts from the strengths of the model. In particular adding phrases such as ‘physical block-spin transformation’ overly complicate the descriptions, when in fact there is a elegance to the simplicity of the model!

- I think the authors should probably include discussion of other models which use multiresolution type structures for long sequence modelling. In particular the authors should definitely discuss multiresnet (https://arxiv.org/abs/2305.01638) which uses a repeated strided convolution at different depths. This is very relevant to the current work and I hope the authors would find interesting. Other possible works (although perhaps less related) include SGConv (https://arxiv.org/pdf/2210.09298) and MRConv (https://arxiv.org/pdf/2408.09453)

- Whilst the originality of the work is good, I'm slightly concerned by the significance of the results. In particular the parameter matching and flop matching experiments don't appear to be in domains where efficiency is essential. Why couldn't we just use a larger model where we don't do weight tying in the DLPFC and Breast Cancer datasets for example? The significance of the results would be increased if the authors presented results in scenarios where the parameter/FLOP counts are critical to the success of the model or its deployment?

- Line 133 C1 - “Following prior work” please provide a citation

- Line 136 “Ensure non-positive eigenvalues” - it’s my understanding that the eigenvalues of the A matrix should be close to 1 in order to learn long-range dependencies (See https://arxiv.org/pdf/2303.06349). Could the authors clarify this point?

- Line 147 “To simulate the physical block-spin transformation” - this needs greater explanation. What is the physical block-spin transformation and why are we simulating it?

- Line 366 “randomly cropped local crops” - I’m not sure this really tests multiresolution capability as the scale of the features is still the same between training and test data. It does test length generalization, but I think it is misleading to suggest it tests scale invariance. A better test for scale invariance would be to downsample all the images to LxL where L<512 and then test on the full resolution images?

---

> ### Author Rebuttal · Authors · 2026-03-29
>
> We thank the reviewer for the careful reading, positive assessment, and constructive suggestions. We are encouraged that the method was found well motivated and empirically strong, especially in terms of FLOP efficiency. We also appreciate the comments on presentation, related work, and the practical significance of the efficiency comparisons. Below we address each point.
>
> **W1**
> We appreciate this suggestion and agree that the RG language can be softened. Our goal is not to claim that DF-Mamba is a literal RG procedure, but rather that the RG viewpoint provides an intuitive and principled motivation for recursive coarse-graining and shared dynamics across scales. In the revision, we will simplify phrases such as “physical block-spin transformation” to avoid overstating the physics analogy and to let the model’s core technical ideas stand out more clearly.
>
> **W2**
> We agree that MultiresNet and related multi-resolution operators such as SGConv and MRConv should be discussed in the revision. We will add these references and clarify the distinction more explicitly. Unlike a standard hierarchical multi-resolution design, our method enforces **shared dynamics across scales** by recursively applying the same state-space operator across recursion depths, together with cross-scale auxiliary constraints that encourage consistency between fine and coarse representations. We will revise the manuscript to make this distinction clearer.
>
> **W3**
> We appreciate this point. We agree that, for datasets such as DLPFC and Breast Cancer ST, efficiency alone is not the primary deployment bottleneck. However, the key contribution of DF-Mamba is not only improved efficiency, but also robust scale generalization under distribution shift, which cannot be achieved by simply increasing model size. Our claim is therefore narrower: the gains are not driven by brute-force scaling. DF-Mamba-ST uses only **1.25M parameters and 4.81G FLOPs**, vs **27.01M and 98.25G for DiffusionST**, while still achieving strong performance. We will revise the wording to make this point precise and avoid overstating efficiency as the main motivation.
>
> **W4**
> We will add the missing citations in the revised manuscript.
> **[1]**"Diagonal state spaces are as effective as structured state spaces."
> **[2]**"On the parameterization and initialization of diagonal state space models."
> **[3]**"Mamba: Linear-time sequence modeling with selective state spaces."
>
> **W5**
> We appreciate the opportunity to clarify this point. The statement refers to the continuous-time generator $A$, which is parameterized to have non-positive eigenvalues for stability. This does not contradict the observation in prior SSM/Mamba work that the discretized transition matrix should be close to identity to preserve long-range dependencies. In fact, after discretization, $\bar{A}=exp(\Delta{A})$, so when the step size $\Delta$ is small and the continuous-time dynamics decay slowly, the eigenvalues of $\bar{A}$ can indeed remain close to 1, corresponding to long memory in discrete time. We will revise the wording to make this continuous-time/discrete-time distinction explicit.
>
> **W6**
> We agree that this sentence currently overstates the physical analogy and does not explain the operation clearly enough. What we actually implement is a simple learnable pairwise coarse-graining operator: neighboring tokens are grouped, projected into a shorter sequence, and supervised with an auxiliary reconstruction term so that coarse representations retain meaningful local information. The block-spin terminology was only intended as an intuition for local aggregation under coarse-graining. We will rewrite this part more directly and reduce the physics-oriented phrasing.
>
> **W7**
> We agree that this setting is more accurately described as zero-shot local-to-global (resolution) extrapolation rather than strict scale invariance. All images are resized to 512×512; training uses only 128×128 or 256×256 local crops, while testing is conducted on the full 512×512 image, unseen during training. Thus, the pixel scale is fixed, but the model must generalize to a much larger field of view and token length without retraining. We will revise the wording accordingly while maintaining that this remains a meaningful and fair stress test, since all baselines are evaluated under the same protocol.
>
> **Key question**
> We performed an ablation on the coarse-graining operator on ETTm2 by replacing the learned pairwise downsampler with max pooling, stride-2 convolution, and spectral low-pass filtering. The results are as follows:
> |  | MSE | MAE |
> |:-|:-:|:-:|
> | learned (ours) | 0.158 | 0.261 |
> | conv | 0.167 | 0.270 |
> | spectral | 0.171 | 0.269 |
> | max | 0.176 | 0.274 |
>
> While convolutional and spectral downsampling are competitive, the learned coarse-graining consistently achieves the best performance. This indicates that the gains arise not only from recursion, but also from the information-preserving coarse-graining design itself.

---

> > ### Author Rebuttal · Reviewer_GwjA · 2026-04-03
> >
> > Thank you to the authors for clarifying my questions, particularly with regard to the non-negative eigenvalues, additional downsampling results and for agreeing to tone down the connections to physics based models which I believe detract from the core contributions of the work.
> >
> > I have a few remaining concerns:
> >
> > **W7**: If the authors agree that their experiment does not actually test for cross-scale generalisation, and the model is always tested on images at the same scale just of different , then I'm slightly concerned by the claim that their model "exhibits zero-shot scale extrapolation". **Could the authors provide a summary of their new position on cross-scale generalization?** This feels like a major selling point of the method and one which is not in actual fact tested. I still feel the paper has other merits worthy of keeping my score a 4 but only on the condition that claims about cross scale generalization are removed and adjusted to reflect what is actually tested for.

---

> > > ### Author Response · Authors · 2026-04-03
> > >
> > > Thank you for this important follow-up. We agree with the reviewer that the histopathology experiment should not be described as a strict test of cross-scale generalization or scale invariance. In that setting, all images are first normalized to a fixed 512×512 resolution; training is performed on local 128×128 or 256×256 crops, while testing is conducted on the full 512×512 image. Therefore, the pathology protocol is more accurately characterized as zero-shot local-to-global / field-of-view extrapolation at fixed pixel scale, rather than strict scale extrapolation.
> > >
> > > Our revised position is therefore the following. We will reserve the term zero-shot cross-scale generalization for settings where the effective scale is genuinely changed between training and testing, such as the native-scale train / coarsened-scale test protocols in time series and spatial transcriptomics. For computational pathology, we will instead use zero-shot local-to-global extrapolation or field-of-view extrapolation, since the challenge is to generalize from limited local context to unseen larger global context without retraining or architectural modification.
> > >
> > > We appreciate the reviewer highlighting this distinction. In the revision, we will remove blanket claims that DF-Mamba demonstrates zero-shot scale extrapolation across all domains, and we will update the abstract, contributions, experimental section, and conclusion so that the terminology matches exactly what is tested.
> > >
> > > Thank you again for your careful reading and constructive feedback.

---

### Decision · Program_Chairs · 2026-04-30

**Decision:**

Accept (regular)

**Comment:**

This paper proposes a multi-resolution SSM model inspired by renormalization group (RG) flow for learning long sequences with complex dynamics. It proposed a novel and technically sound RG-inspired formulation for a well-motivated problem; competitive empirical performance across three tasks with well-designed experiments.

**Rebuttal summary**
- The reviewers and authors agree that the theoretical connection to RG is loose and needs reposition as a motivation.
- The scale-invariance experiments is questionable in the initial submission (same-resolution train/test, missing two tasks). Authors clarified experiments as supporting local-to-global generalization, and provided additional cross-scale evaluations (train/test on different scales), as well as additional baseline comparisons and ablations on downsampling. These additional results largely resolving the key concerns.

**Recommandation**
Accept. A sound and novel contribution to long-sequence modeling for complex dynamics. With the revised framing, scope clarification and additional experiments, it is a valuable addition to the conference.